# Nanoscale exchange-bias magnetic tunnel junctions enabled memristive synapse and leaky-integrate-fire neuron for neuromorphic computing

Zanhong Chen[1,4], Dehang Zhu[1,4], Ao Du[1,4], Yuzhang Shi[1,4], Wenlong Cai [1,2] ✉, Zixi Wang [1], Yuqi Duan[1], Shiyang Lu[1], Kaihua Cao[1], He Zhang[1], Deming Zhang[1], Hongxi Liu [3], Kewen Shi [1,2] ✉ & Weisheng Zhao [1,2] ✉

Neuromorphic computing implemented by spintronic memories offers computational advantages, including in-memory computing capability, high energy efficiency, and near-unlimited endurance. However, their basic units, magnetic tunnel junctions (MTJs), face inherent challenges in emulating analog synapses and spiking neurons due to their bistable resistance states and lack of bio-realistic switching dynamics. Here, we experimentally demonstrate on-chip co-integrated memristive synapse and leaky-integrate-fire (LIF) neuron designed with nanoscale exchange-bias MTJs (EB-MTJs). By exploiting the spatial distribution of antiferromagnets and its current-dependent modulation, we achieve stable continuous multi-state synaptic behavior with spike-timing-dependent-plasticity (STDP) characteristics in compact (~100 nm) EB-MTJs. Furthermore, time-resolved measurements reveal that EB-MTJs can be progressively programmed by 0.4 ns pulses, emulating LIF neuronal dynamics with high operational bandwidth in the gigahertz range. Finally, we construct a convolutional spiking neural network based on EB-MTJs and achieve 96% accuracy in gesture recognition via a hybrid backpropagation-STDP algorithm, highlighting their potential in neuromorphic computing.

Neuromorphic computing is an emerging computational paradigm inspired by biological neural systems[1,2]. By emulating the behavior of neurons and synapses, it promises energy-efficient, high-speed, and scalable information processing with inherent fault tolerance characteristics well-suited for intelligent tasks[3–5]. Among neuromorphic models, spiking neural networks (SNNs) have gained increasing attention due to their event-driven sparsity, temporal coding, and high biological plausibility, which enable improved energy efficiency and information density compared to traditional neural networks[6–9].

However, hardware realization of SNNs remains challenging. The mainstream hardware for current network implementation still relies on conventional von Neumann architectures implemented in CPU/GPU-based systems, where memory and computation are physically separated. This approach deviates from the intrinsic co-location of memory and processing in spiking neurons and synapses, thereby undermining the structural advantage of SNNs and resulting in significant energy and latency overheads[10,11]. Furthermore, CMOS-based spiking neurons and synapses require numerous transistors and

[1]Fert Beijing Institute, School of Integrated Circuit Science and Engineering, Beihang University, Beijing, China. [2]State Key Laboratory of Spintronics, Hangzhou International Innovation Institute, Beihang University, Hangzhou, China. [3]Truth Memory Tech. Corporation, Beijing, China. [4]These authors contributed equally: Zanhong Chen, Dehang Zhu, Ao Du, Yuzhang Shi. ✉e-mail: caiwenlong1993@buaa.edu.cn; shikewen@buaa.edu.cn; weisheng.zhao@buaa.edu.cn

capacitors, resulting in non-negligible area cost and power consumption[12,13].

To address inherent limitations of traditional semiconductors, emerging non-volatile memory technologies, including resistive random access memory (RRAM)[14,15], phase-change RAM (PC-RAM)[16], ferroelectric RAM (FeRAM)[17] and magnetoresistive RAM (MRAM)[18], have been extensively investigated. These devices offer in-memory computing capabilities, positioning them as promising candidates for neuromorphic computing[19]. Among them, MRAM stands out due to its ultrafast switching, low power consumption, and high endurance[20–23]. Moreover, their memristive characteristics and rich switching dynamics, driven by spin-transfer torque (STT) and spin-orbit torque (SOT), facilitate compact emulation of synaptic and neuronal dynamics[24–28], showing great potential for implementing biologically inspired computing systems.

Extensive efforts have been devoted to realizing synaptic and spiking neuronal functionalities using magnetic tunnel junctions (MTJs), the core cells of MRAM. For synapses, early implementations relied on binary MTJ arrays, where synaptic weights were encoded through array-level configurations, resulting in significant area and energy overheads[29]. Recent advances have focused on multi-state synapses enabled by domain-wall (DW) devices, including geometry-engineered tracks[30], Dzyaloshinskii–Moriya interaction-driven DW motion[22], and chiral DW configurations[24]. Skyrmions have also been explored in MTJs for compact multi-level devices[31,32]. Despite enabling discrete resistance modulation, these approaches either exhibit a limited number of stable resistance states or require a large area, making the realization of compact memristive synapse elements highly challenging. Meanwhile, various exploratory MTJ designs have been proposed to emulate spiking neurons mainly by magnetic precession accumulation[33,34] or DW motion[35,36]. However, they suffer from

a lack of experimental validation in the former case, and from speed and scalability challenges in the latter. More importantly, achieving monolithic integration of compact memristive synapses and spiking neurons within a single platform remains a key challenge toward hardware-efficient neuromorphic systems.

Herein, we successfully demonstrate nanoscale memristive synapses and spiking neuron devices with exchange-bias (EB) MTJs, whose free layer is also pinned via exchange bias, achieving monolithic integration by a unified film stack. By leveraging the intrinsic stability of antiferromagnets and their spatially distributed grain sizes, we achieve stable continuous multi-state behavior in compact elliptical devices (~100 nm), enabling synaptic functions such as linear weight modulation and spike-timing-dependent-plasticity (STDP). Exploiting the interplay between SOT and thermally assisted accumulation, we experimentally validate leaky-integrate-fire (LIF) neuronal dynamics with sub-nanosecond response time. Moreover, by integrating both device types, we then implement a fully spintronic convolutional spiking neural network (CSNN) achieving 96% accuracy in gesture recognition. These results highlight EB-MTJ as a promising platform for compact and high-speed brain-inspired computing.

## Results

### Neuromorphic device concepts and basic properties

Figure 1a illustrates the implementation of both LIF neurons and STDP synapses in EB-MTJ devices. Despite their distinct functional roles, both types of units share an identical film stack, which is composed of Si/SiO$_2$ substrate/Pt (8)/Ir$_{20}$Mn$_{80}$ (4.5)/Co$_{20}$Fe$_{60}$B$_{20}$ (1.9)/MgO (1.5)/Co$_{20}$Fe$_{60}$B$_{20}$ (2.4)/Ru (0.8)/CoFe$_{30}$ (2)/Ir$_{20}$Mn$_{80}$ (7.5) (thickness in nanometers). Here, we employ an Fe-rich CoFeB alloy as the ferromagnetic layer, which is known to provide high tunneling magnetoresistance (TMR) and excellent thermal stability in MgO-based

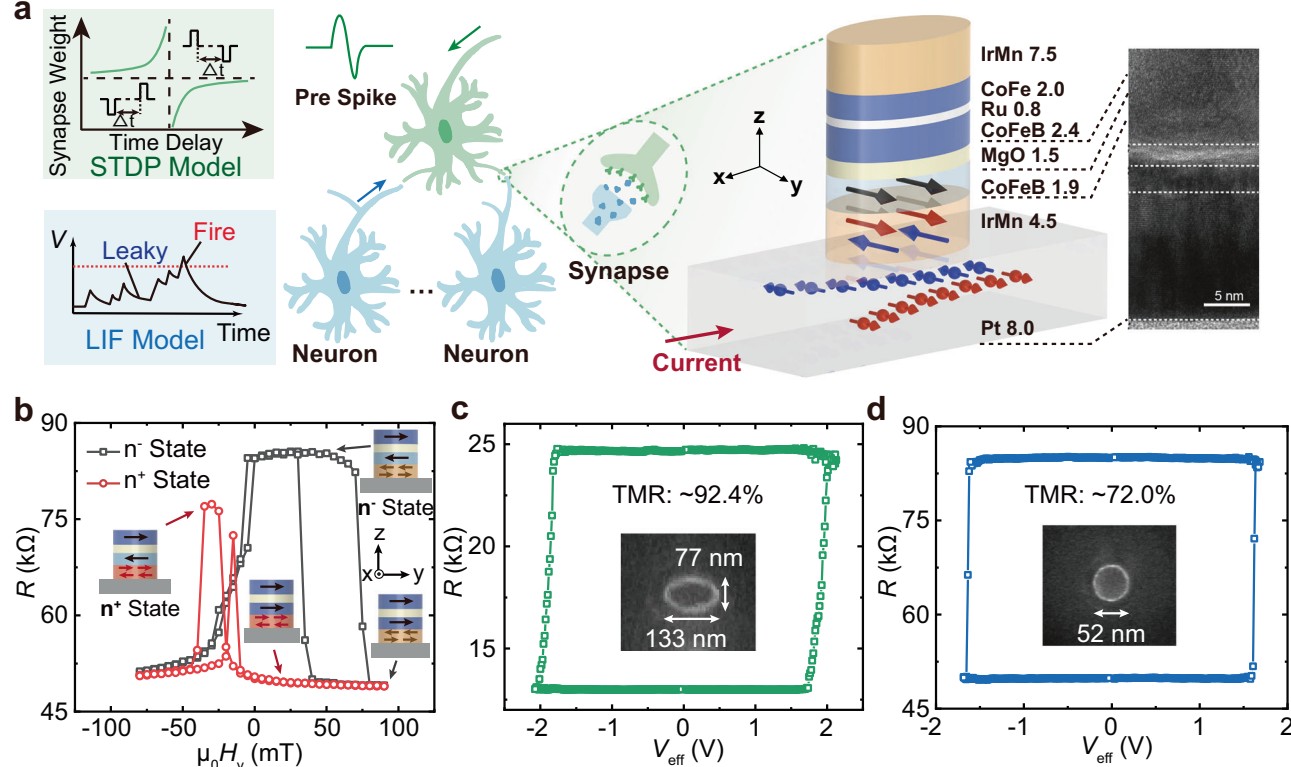

**Fig. 1 | Structure and fundamental properties of artificial neuron and synapse.** **a** The device functionalities for spiking neural networks and schematic of the EB-MTJs stack structure with cross-sectional transmission electron microscopy (TEM) image. **b** Magnetic hysteresis loops of the device with the antiferromagnetic layer preset to the **n$^+$** and **n$^-$** states, corresponding to the P state and AP state under zero field. **c,d** SOT switching curves of the circular and elliptical devices under 0.4 ns pulses, with their corresponding SEM images. The TMR ratios of the two devices reach 72.0% and 92.4%, respectively.

MTJs[37]. The top IrMn antiferromagnet (AFM) layer, together with the RKKY-coupled SAF, stabilizes the magnetization of the reference layer. In addition, the bottom IrMn layer stabilizes the magnetization of CoFeB via exchange bias, whereas the bottom Pt electrode acts as a spin Hall source for SOT-induced switching and as a source of an Oersted field[38–40]. Hereafter, IrMn refers specifically to the bottom IrMn layer unless otherwise noted. The full stack is annealed at 300 °C for one hour under a 1 T magnetic field along the -y axis, followed by field cooling. This procedure aligns the magnetization of the device along the -y direction, consistent with the spin polarization generated by the SOT current flowing along the x direction in the underlying Pt electrode. Cross-sectional TEM reveals continuous interfaces across the multilayer stack and clear crystalline features in the MgO barrier and adjacent CoFeB layers, which are known to play a key role in achieving high TMR in MgO-based MTJs.

MTJs with diverse aspect ratios and lateral dimensions are fabricated, exhibiting TMR values exceeding 70%, as shown in Fig. 1b. The hysteresis loop of a circular device exhibits a single stable resistance state at zero field due to strong exchange bias, ensuring robust data retention. Here, we define the $\mathbf{n^-}$ state (black curve) as the configuration in which the EB field is oriented along the −y direction. At zero external magnetic field, the magnetization of the CoFeB free layer is stabilized along −y by the EB field, while the reference layer magnetization remains along +y, corresponding to the antiparallel (AP) state. Conversely, the $\mathbf{n^+}$ state (red curve) corresponds to an EB field oriented along +y, for which the CoFeB free-layer magnetization aligns along +y at zero field, resulting in the parallel (P) state. Notably, the $\mathbf{n^-}$ and $\mathbf{n^+}$ configurations are preset electrically by SOT current prior to the field sweep in Fig. 1b. Moreover, the resistance state of the EB-MTJ can be modified by applying write voltage pulses through the bottom electrode, with pulse widths down to 0.4 ns as shown in Fig. 1c and d. The detailed pulse shapes and measurement procedures are described in the Methods and in Supplementary Note 1 and 2.

Interestingly, devices with different geometries and sizes exhibit markedly different resistance responses after voltage-pulse writing. Elliptical devices with a size of 77 nm × 133 nm exhibit robust memristive behavior after voltage-pulse writing, whereas circular devices with a smaller size of 52 nm demonstrate sharp binary transitions, making them suitable for synaptic and neuronal roles, respectively. This distinction can be attributed to the reversal mode of the free layer: sufficiently small devices favor quasi-single-domain switching, while devices with larger lateral sizes and higher aspect ratios support non-uniform reversal involving relatively stable magnetic domain configurations, enabling multiple intermediate resistance states (see Supplementary Note 3 for details). The formation and stability of these multi-state configurations are discussed in the following section. To ensure accurate quantification of the switching voltage, we define the effective SOT voltage $V_{eff}$ as the voltage drop across the constricted bottom electrode region directly beneath the MTJ. Details of this calculation are provided in Supplementary Note 4. These contrasting switching behaviors demonstrate the device's versatility in supporting both spiking and analog-like operations. Such tunable switching dynamics are essential prerequisites for integrated spintronic neuromorphic architectures, which will be further explored in subsequent sections.

## Nanoscale synaptic device with spike timing plasticity

To explore the synaptic functionalities enabled by the EB-MTJ structure, we investigate the analog-like multi-level resistance states exhibited by the elliptical devices. As shown in Fig. 2a, we applied 0.4 ns SOT pulses with varying current densities to the device, achieving over 25 discrete resistance states. Further measurements confirm that multi-level operation is maintained up to pulse widths of 5 ns (typically >20 levels), with the resistance levels remaining clearly distinguishable, enabling reliable multi-state operation (details are

provided in Supplementary Note 5 and 6). Meanwhile, Fig. 2b and c demonstrate the thermal and magnetic field stability of the device, respectively. Specifically, the EB-MTJ exhibits a thermal stability factor exceeding 100, corresponding to a retention time exceeding 10 years, as detailed in Supplementary Note 7. Furthermore, Fig. 2c shows that although the device resistance varies under the application of a perpendicular magnetic field, it fully recovers to its original value after field removal, even at fields up to 2 T (details are provided in Supplementary Note 8). This field-removal invariance, together with the robust thermal stability, confirms that the EB-MTJ devices maintain stable and discrete memristive states against thermal and magnetic perturbations.

The memristive behavior observed in synaptic devices could originate from the magnetization pinning at the AFM/FM interface[38,41,42], which stabilizes multiple intermediate domain configurations during switching. Therefore, the grain size distributions were quantified in a 6 nm-thick $Ir_{20}Mn_{80}$ single layer and in a sandwiched structure of $Co_{20}Fe_{60}B_{20}$ (4 nm)/$Ir_{20}Mn_{80}$ (6 nm)/$Co_{20}Fe_{60}B_{20}$ (4 nm)[43,44]. As shown in Fig. 2d, the grain sizes follow a log-normal distribution[45], based on data from over 100 grains. The grain-size distribution can be described by a log-normal function, which can be expressed as[46]

$$f(v)dv = \frac{1}{\sqrt{2\pi}\sigma v} \exp\left[ -\frac{(\ln(v) - \mu)^2}{2\sigma^2} \right] dv \qquad (1)$$

where $v$ is the grain size and $\ln(v)$ is normally distributed with a mean value $\mu$ and a standard deviation $\sigma$. The fit parameters are $\mu = 2.04$ and $\sigma = 0.190$ for the $Ir_{20}Mn_{80}$ single layer (yellow), and $\mu = 2.14$ and $\sigma = 0.199$ for the sandwiched structure (green), corresponding to median grain sizes of 7.71 nm and 8.47 nm, respectively. Since CoFeB is amorphous (see Supplementary Note 9), the circular grains in the sandwiched structure suggest that the grain structure of CoFeB is influenced by the IrMn domains, resulting in enlarged and more circular grains. In such cases, the magnetic moments of CoFeB domains are locally constrained by the underlying IrMn grains and follow their orientation.

To further clarify the current-induced mechanism for writing intermediate resistance states, the magnetic domain configurations of both AFM and FM layers under excitation by input voltages are investigated by micromagnetic simulation on OOMMF[47] (details in Methods). As the input voltage increases, a change in device resistance is observed as shown in Fig. 2e, indicating that partial domain switching occurs in the ferromagnetic free layer.

This partial switching originates from the heterogeneous exchange-bias landscape imposed by the AFM layer. Antiferromagnets such as $Ir_{20}Mn_{80}$ exhibit a distribution of grain sizes, giving rise to a broad distribution of energy barriers among individual grains. Larger antiferromagnetic grains possess sufficient anisotropy energy to remain magnetically stable at room temperature, with their blocking temperature determined by the thermal activation condition:

$$T_B(v) = \frac{K_{AF} v}{k_B \ln(\tau_m / \tau_0)} \qquad (2)$$

where $K_{AF}$ is the anisotropy energy density of the antiferromagnet, $k_B$ is the Boltzmann constant, $\tau_m$ is the measurement time, and $\tau_0$ is the attempt time. During a 0.4 ns SOT pulse, AFM grains with smaller lateral dimensions undergo pronounced precessional dynamics driven by the combined effects of Joule heating and SOT, leading to their dynamical decoupling from the neighboring FM domains and the local loss of AFM/FM interfacial exchange coupling[40], as shown in the upper panels (transient dynamics) of Fig. 2e. In these regions, the local FM domains can be reversed by the combined action of the Oersted field and SOT. The coexistence of such locally reversed domains leads to an intermediate magnetic configuration in which the FM layer adopts a

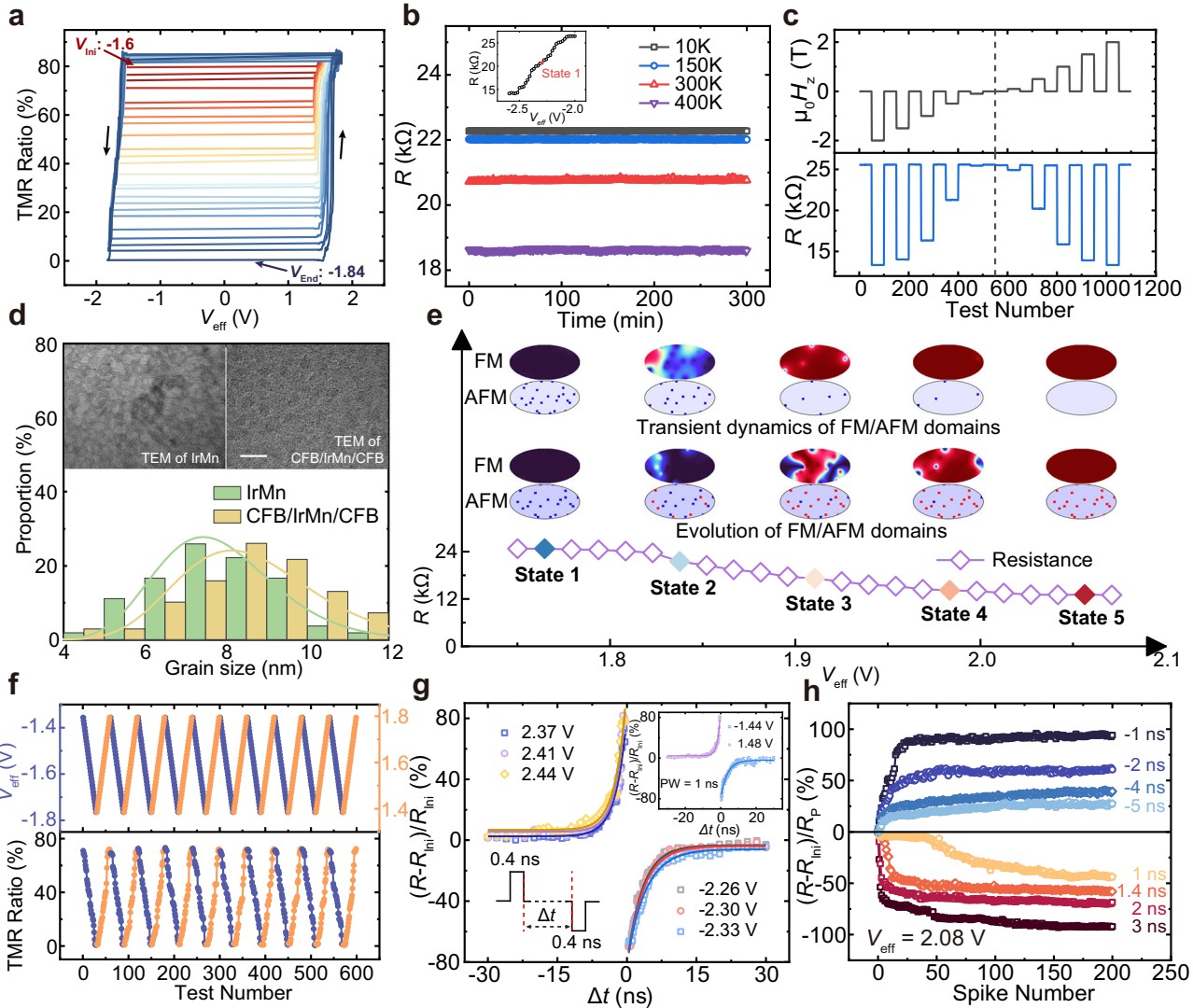

**Fig. 2 | Synaptic device exhibiting STDP characteristics. a** Analog memristive switching loops obtained by applying 0.4 ns SOT pulses with gradually increasing current amplitudes, exhibiting more than 25 stable resistance states. **b** Time stability of intermediate resistance states, demonstrating robust data retention. The inset illustrates the intermediate resistance state of the device during the retention test. **c** Magnetic immunity of the device under an external magnetic field applied along the y direction. **d** Grain size distributions in the $Ir_{20}Mn_{80}$ and the $Co_{20}Fe_{60}B_{20}/Ir_{20}Mn_{80}/Co_{20}Fe_{60}B_{20}$ multilayer structure, illustrating the pinning effect of the AFM layer on the adjacent ferromagnet (FM) layer. Scale bar: 10 nm. **e** Micromagnetic simulation showing the evolution of AFM and FM domains, with device resistance during thermally assisted SOT switching, the blue regions denote unswitched magnetic domains, whereas red regions indicate switched ones. **f** Linear dependence of resistance state on the amplitude of SOT pulses. **g** STDP behavior induced by paired pre- and post-synaptic SOT pulses with varying Δ$t$. **h** The modulation of 200 consecutive 0.4 ns pulses with different time intervals on the device resistance (synaptic weight).

multidomain state. The AFM grains that undergo dynamical decoupling are characterized by[48]

$$\nu < \nu_c = \frac{k_B T \ln(\tau_m/\tau_0)}{K_{AF}\left(1 - \frac{V}{V_{c0}}\right)} \qquad (3)$$

where $\nu_c$ is the critical grain volume at a given temperature, $V_{c0}$ is the threshold voltage at zero temperature and $V$ denotes the applied voltage.

Upon current withdrawal, as the temperature decreases, the AFM domains beneath the already reversed FM domains re-couple to the FM domains and re-establish a local exchange bias that follows the switched FM magnetization. As a consequence, the FM multidomain configuration becomes strongly pinned by the reformed exchange bias and is rendered highly stable, as shown in the middle panels of

Fig. 2e. This grain-resolved re-coupling process results in a spatially mixed exchange-bias landscape.

As a result, the exchange-bias reconfiguration and the resulting spatially inhomogeneous exchange-bias landscape described above act to stabilize the multidomain configuration of the FM layer. This stabilization directly manifests at the device level as distinct intermediate resistance states of the MTJ, as captured in the bottom panel of Fig. 2e. As the SOT current amplitude increases, a larger fraction of AFM grains undergoes decoupling and reorientation, progressively enhancing the degree of mixed exchange bias and leading to a gradual transition of the device from the AP to the P state (States 2–5). This grain-selective pinning mechanism is fundamentally distinct from conventional SOT-MTJs with purely ferromagnetic free layers, where the absence of such exchange-bias reconfiguration leads to binary switching behavior.

The thermally assisted domain evolution mechanism enables analog switching in the device, facilitating effective emulation of

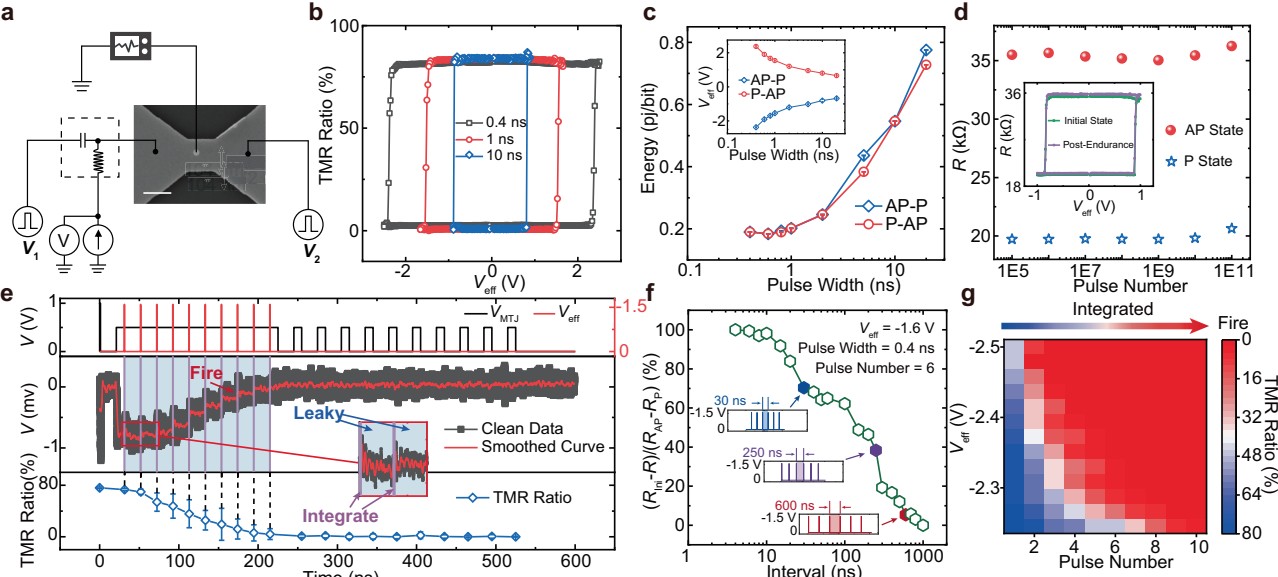

**Fig. 3 | Neuronal device exhibiting LIF characteristics. a** Experimental diagram for time-resolved characterization of the neuron device. Scale bar: 500 nm. **b** Switching curves under different pulse widths. **c** Energy consumption required for writing under different pulse widths, with error bars representing the standard deviation (SD). The inset shows the critical switching voltage of the device under different pulse widths. **d** Resistance of the device in the AP (P) state as a function of write pulse number up to 1E11. The inset shows that the switching loops exhibit negligible change before and after the endurance test, demonstrating the excellent endurance of the device. **e** Time-resolved resistance evolution under repeated sub-threshold pulses, demonstrating thermal integration and leaky relaxation, where the blue curves show the device resistance values under different pulse numbers with error bars representing the SD. **f** Dependence of integration and leakage dynamics on the inter-pulse interval. **g** Integrate-and-fire states of the device under various pulse numbers and current densities.

synaptic weight plasticity. STDP enables unsupervised adaptation of synaptic weights based solely on spike timing, making it a key mechanism for energy-efficient and hardware-compatible neuromorphic learning systems[49]. To validate the capability of the memristive MTJs to emulate the STDP model, we applied two 0.4 ns SOT pulses with opposite current direction and a temporal interval ($\Delta t$) to the device, simulating the pre- and post-synaptic neuronal spikes. The first pulse induces local heating, thereby lowering the energy barrier, while the second pulse performs the effective state update. As shown in Fig. 2g, the magnitude of synaptic weight change increases as $|\Delta t|$ decreases, consistent with the biological observation that synapses between frequently co-active neurons experience greater plasticity.

Furthermore, the synaptic device undergoes LTD and LTP under repeated 0.4 ns positive and negative pulses, respectively (Fig. 2h), demonstrating stable and controllable long-term plasticity suitable for emulating adjustable synaptic weights in neuromorphic circuits. As shown in Fig. 2f, by applying a series of SOT pulses with gradually increasing amplitude, the synaptic device exhibits a linear resistance response to the pulse voltage. Notably, this behavior is reproduced over ten consecutive programming cycles, demonstrating the stability and high reproducibility of the analog switching characteristics, which is essential for applications in neuromorphic computing. Beyond standard STDP, the thermal-assisted writing mechanism can also implement other plasticity rules, such as BCM, which relies on the correlation between the average activity of the pre- and post-synaptic neurons. The underlying principle and experimental implementation of BCM using EB-MTJs are described in Supplementary Note 10. These results demonstrate that analog synapses based on compact EB-MTJs can serve as a robust foundation for implementing neuromorphic functionalities in hardware-based SNNs.

### Neuronal device with leaky integrate and fire dynamics

Importantly, supporting both synaptic and neuronal functionalities within a single spintronic platform is crucial for scalable and energy-efficient neuromorphic systems. We next investigate the LIF neuron dynamics in circular EB-MTJ devices. Although neuronal behavior was verified in a 52 nm device, a 104 nm device with lower resistance was employed here to enable clearer time-resolved oscilloscope measurements. As shown in Fig. 3a, the output channels of an arbitrary waveform generator were connected to the bottom electrodes of a 104 nm EB-MTJ (see Methods), while the top electrode was grounded through an oscilloscope for real-time voltage monitoring. Figure 3b demonstrates deterministic switching under SOT pulses as short as 0.4 ns, showcasing the potential for ultrafast neuronal operation. The effective energy consumption, estimated for a 104 nm × 104 nm bottom electrode which is sufficient for an EB-MTJ neuron, is approximately 190 fJ per switching event, significantly lower than that of conventional CMOS neurons[50]. These results demonstrate the ultrafast and low-power characteristics of the EB-MTJ, offering advantages for hardware-efficient spiking neuron implementations.

To emulate LIF neuronal behaviors, we define the high-resistance state as the resting state and the low-resistance state as the firing state. The neuronal devices follow the same thermally assisted switching principle as synaptic devices, while the switching strategy is adapted for neuronal operation. A sequence of sub-threshold SOT pulses, each individually insufficient to switch the device, cumulatively lowers the energy barrier via Joule heating, effectively realizing the integrate phase. Once the barrier falls below a critical threshold, a subsequent pulse triggers a state transition, representing the firing event. In the absence of further stimulation, the barrier gradually recovers through thermal relaxation, leading to the leaky dynamics.

This mechanism was validated through time-resolved resistance measurements. In the experiment shown in Fig. 3e (upper panel), the device was stimulated by a train of ten 0.4 ns SOT pulses at 20 ns intervals (exact pulse shapes are provided in Supplementary Note 1), while a constant 0.5 V bias ($V_{test}$) was applied for continuous resistance monitoring. As shown in the middle panel, the device resistance decreases during the writing pulses (purple-shaded regions), indicating thermal accumulation and input integration. During the read-only intervals (blue-shaded regions), resistance gradually recovers due to

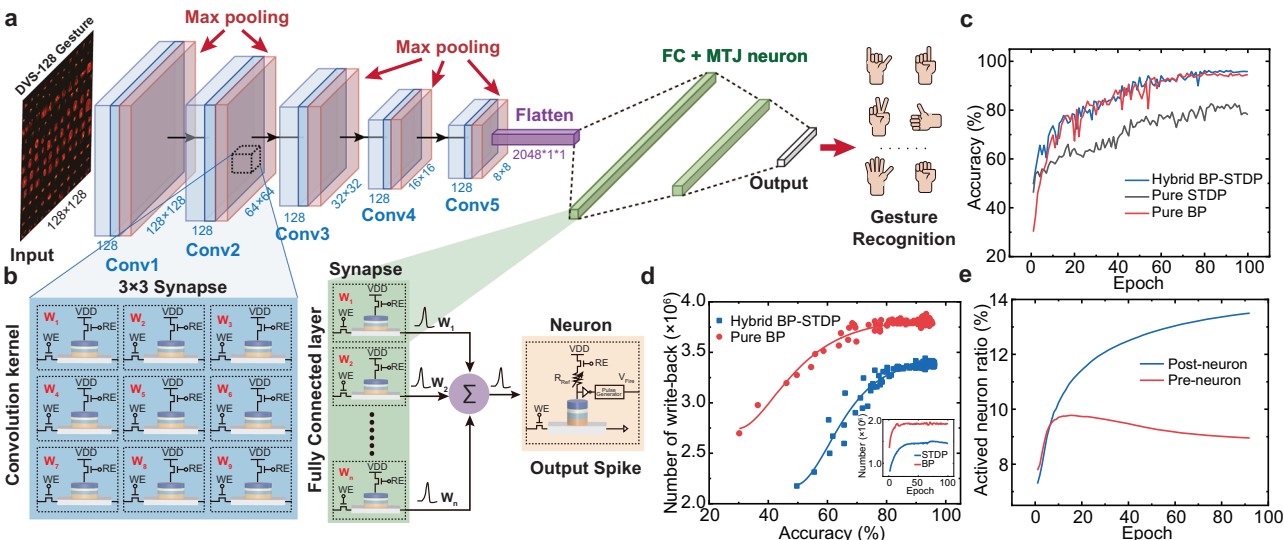

**Fig. 4 | Structure and Results of SNN Network Simulation. a** Topology of the five-convolution/two-fully-connected (5-Conv/2-FC) network employed in the simulation. The input event frames are from the publicly available DVS128 Gesture dataset[52]. **b** Schematics of a 3 × 3 convolutional kernel, a fully connected synapse, and a spiking neuron based on EB-MTJs. **c** Test accuracy versus epoch for hybrid BP-STDP (blue), pure STDP (black) and pure BP (red). **d** Comparison of write-back numbers for hybrid BP-STDP and BP schemes versus accuracy. Inset shows the comparison of write-back numbers for STDP and BP during the training process. **e** Fraction of neurons active during training and inference.

heat dissipation, reflecting the leaky behavior. Continued pulsing ultimately switches the device from the $n^-$ to the $n^+$ state, marking the firing event. Once the energy barrier drops below a critical threshold, a subsequent pulse triggers the firing event, which can be read out via appropriate sensing circuitry (see Methods for details). The lower panel shows the averaged resistance after different numbers of pulses over 20 repetitions, clearly revealing increased switching probability with pulse accumulation. Figure 3f and g further demonstrate that shorter pulse intervals, larger pulse counts, and higher amplitudes all increase the firing probability, closely mimicking the response of biological neurons to more frequent or stronger inputs. These results demonstrate that EB-MTJs can reliably replicate key features of LIF neurons using the same thermally assisted switching mechanism as for synaptic weight modulation. Crucially, both synaptic and neuronal functionalities are achieved within the same film stack, enabling compact and monolithically integrated spintronic SNNs.

**Convolutional spiking network with magnetic tunnel junction**
To evaluate the system-level capabilities of the proposed EB-MTJ LIF and STDP primitives, we implemented a 5-Conv/2-FC CSNN based on SpikingJelly[51] and benchmarked it on the DVS-Gesture dataset[52]. The asynchronous 128 × 128 event stream of each gesture was discretized into eight time bins and formatted as a tensor of shape T × C × H × W = 8 × 2 × 128 × 128. The five convolutional blocks (Conv-BN-ReLU-MaxPool) compressed the representation to 4 × 4 × 128. The resulting feature map was flattened and forwarded to two hidden fully connected layers, followed by an 11-way classifier. All neurons were modeled using parameters derived from the thermally assisted switching dynamics of EB-MTJs, with the mapping of device-level parameters to the neuron model described in the Methods, and the detailed simulation procedures described in Supplementary Note 11.

Due to the vanishing-plasticity issue in deep SNNs trained solely with STDP, we adopted a hybrid training strategy. Specifically, surrogate-gradient back-propagation (BP) was used to optimize all convolutional layers and the first fully connected layer, while the second fully connected layer was fine-tuned online using a bi-exponential STDP rule calibrated from EB-MTJ measurements. This hybrid BP-STDP approach allows convolutional layers to extract robust spatio-temporal features, while enabling the output layer to form class-specific associations in a hardware-compatible, event-driven manner. To prevent update conflicts and mitigate gradient mismatch, we decouple the learning paths by applying BP only to the BP-trained parameter group with surrogate gradients for spikes and applying STDP only to the STDP-trained layer via a separate local update/optimizer after clearing BP gradients on that layer.

As shown in Fig. 4c, the hybrid scheme achieved a test accuracy over 96% after 100 training epochs, outperforming pure BP and pure STDP by 1.5 and 11.7 percentage points, respectively. After accounting for device-to-device variability, the proposed scheme maintains an accuracy above 94%, as described in Supplementary Note 12. These results confirm that BP alleviates the vanishing-plasticity problem, whereas STDP introduces biologically inspired local learning. Moreover, the local update nature of STDP reduces the number of synaptic updates by ~24% compared to pure BP (inset of Fig. 4d). As a result, when the classification accuracy exceeds 96%, our hybrid BP-STDP scheme reduces the total number of updates by ~11% compared to pure BP, demonstrating improved write efficiency as shown in Fig. 4d. Owing to the sparse input spike trains and the cumulative behavior of EB-MTJ neurons, fewer than 14% of neurons needed to fire during training as shown in Fig. 4e, resulting in reduced energy consumption. Compared to conventional artificial neural networks in which all neurons are typically activated, our SNN delivers comparable accuracy with enhanced energy efficiency.

## Discussion
We present nanoscale EB-MTJs exhibiting both neuronal and synaptic functionalities within the same film stack, enabling ultrafast and scalable SNN hardware. We demonstrate memristive switching in compact synaptic devices (~100 nm), achieving over 25 stable resistance states and robust STDP. Micromagnetic simulations reveal that the spatial distribution and current-induced reconfiguration of AFM grains generate heterogeneous switching conditions, stabilizing multiple intermediate states, and suggest that future improvements in capacity could be realized through antiferromagnetic microstructure engineering—by reducing the effective grain size and tailoring the blocking-temperature distribution, even finer and more numerous

intermediate states may be stabilized. Furthermore, reliable switching is realized in smaller circular EB-MTJs with 0.4 ns current pulses, where Joule heating modulates the energy barrier to realize LIF dynamics, with an low power consumption of 190 fJ per switching event. Leveraging these two spintronic building blocks, we implement an all-EB-MTJ CSNN, achieving over 96% accuracy in a gesture recognition task. A hybrid BP-STDP training scheme is employed, attaining higher accuracy while leveraging the 24% reduction in synaptic update operations offered by STDP over pure BP. Meanwhile, the proportion of active neurons remains below 14%, further reducing power consumption. These results demonstrate the promising platform of EB-MTJs for compact, ultrafast and energy-efficient all-spin neuromorphic hardware.

## Methods

### Film deposition and device fabrication

The MTJ film stacks were deposited on a thermally oxidized silicon substrate by direct-current (DC)/radio frequency (RF) magnetron sputtering. The stacks were annealed at 300 °C in a vacuum for 1 h in a magnetic field of 1 T along the -y direction and were then cooled from 300 °C to room temperature under the same field. Subsequently, the bottom electrode pattern was defined via electron beam lithography (EBL), followed by argon ion beam etching with secondary ion mass spectrometry for endpoint detection. MTJ nanopillars were then patterned by EBL to form circular or elliptical shapes and etched using ion milling to define the pillar structure. After etching, a $Si_3N_4$ insulating layer was deposited by plasma-enhanced chemical vapor deposition. A lift-off process was then performed to expose the top surfaces of the MTJ and via regions. Finally, the top electrodes were defined by ultraviolet photolithography and formed by e-beam evaporation of Ti (20 nm) and Au (100 nm), followed by lift-off.

### Measurements

The schematic of the electrical measurement setup is shown in Fig. 3a. A bias tee was used to separate the DC and RF paths, allowing for both static and sub-nanosecond dynamic electrical characterizations. For DC measurements, a Keithley 6221 A current source and a Keithley 2182 A voltmeter were employed to probe the resistance state of the device after programming. Magnetic hysteresis loops were obtained by combining this setup with a GMW Associates 5201 Projected Field Electromagnet. For high-speed pulse measurements, a Keysight M8190A arbitrary waveform generator with a 0.2 ns rise time was used in conjunction with an external amplifier to produce two synchronized voltage pulses, $V_1$ and $V_2$, with opposite polarity and amplitudes up to 8.5 V. These pulses were applied to both ends of the Pt bottom electrode, while the top electrode of the MTJ was connected to a Keysight DSOV334A oscilloscope and grounded. By tuning the amplitudes of $V_1$ and $V_2$, the applied SOT voltage and the MTJ bias voltage can be independently controlled using the following relations: $V_{SOT} = V_1 - V_2$, $V_{MTJ} = (V_1 + V_2)/2$. During current-induced SOT switching tests, we set $V_1 = -V_2$; after the pulse voltages were removed and the device resistance stabilized, the resistance was measured using a 1 μA DC current. Except for the retention tests performed under varying temperatures, all resistance measurements were conducted at room temperature (300 K). During time-resolved measurements, $V_{MTJ}$ was fixed at 0.5 V to allow readout of the voltage signal on the oscilloscope.

Due to the hourglass-shaped geometry of the Pt bottom electrode, with the MTJ located in the narrow central region, we define the effective SOT voltage as the voltage drop across this constricted section. Detailed electrode geometry and resistance segmentation are illustrated in Supplementary Note 4. Based on the bottom electrode layout, we computed the number of square resistances in the central region ($N_{eff}$) and in the entire bottom electrode ($N_{MBE}$). The effective

voltage applied across the MTJ region was then calculated by: $V_{eff} = (V_{SOT} \cdot N_{eff})/N_{MBE}$. This calibration ensures accurate estimation of the local current density during switching experiments.

For magnetoresistance measurements under large magnetic fields, we used a Lake Shore CRX-VF probe station, capable of applying in-plane or out-of-plane magnetic fields up to ±2.5 T. Resistance measurements were again performed using the Keithley 6221 A/2182 A setup.

### Micromagnetic analysis of memristive dynamics in device

All micromagnetic simulations in this study were performed using the OOMMF software (version 2.0b0). Custom Tcl scripts were employed to define and manipulate spatially non-uniform external fields, enabling the simultaneous incorporation of thermally activated pinning mechanisms, Oersted fields, and spin–orbit torque driving terms. The simulated structure was an elliptical nanodisk with in-plane dimensions of 160 nm × 80 nm and a thickness of 1.4 nm. A finite difference discretization scheme was used, with a mesh size of 1 nm × 1 nm × 1.4 nm. This resolution is smaller than the typical exchange length and sufficient to resolve domain walls and other non-uniform magnetization features.

The material parameters correspond to those of typical CoFeB thin films: the saturation magnetization was set to $M_s = 1.3 \times 10^6$ A·m$^{-1}$ to define the magnetization magnitude; the exchange stiffness $A = 16$ pJ·m$^{-1}$ governs spin alignment rigidity and domain wall width; the uniaxial perpendicular anisotropy constant $K_u = 2.9 \times 10^4$ J·m$^{-3}$ ensures in-plane magnetization preference; and the film thickness was 1.4 nm, consistent with commonly used ultrathin experimental structures.

The total effective magnetic field included contributions from exchange interaction, uniaxial anisotropy, demagnetizing field (dipolar interaction), Oersted field, and SOT. Simulations were initialized from a uniform magnetization state pointing along the +y direction. Time evolution was controlled using the SpinXferEvolve module, which supports spin-torque effects. A damping constant of $\alpha = 0.1$ was used, and the system was evolved using a fourth-order Runge–Kutta integration scheme with a maximum time step of 2 ps. Each simulation lasted for 2 ns.

The spin–orbit torque term was incorporated via the SpinXfer-Evolve module and defined by the spin polarization rate, spin polarization direction, and normalized current density. The SOT acted along the −y direction. At the same time, a current-induced Oersted field was applied through the script, also oriented along the −y axis and spatially confined to the film shape by a geometric mask. These two current-driven torques acted simultaneously during the simulation, jointly modulating the magnetization switching behavior. The relevant parameters were: spin polarization rate $P = 0.6$ and spin–orbit torque efficiency $\theta = 0.15$.

The thermally activated pinning effect was modeled using a virtual grain structure. The magnetic region was assumed to consist of grains with a log-normal distribution of in-plane diameters, with a mean of 5 nm and a standard deviation of 0.19. The volume of each grain was estimated from its size and used to calculate the blocking temperature according to the relation:

$$T_B = K_{AF} \times V/(k_B \times \Delta) \tag{4}$$

where $K_{AF} = 1 \times 10^6$ J·m$^{-3}$, $\Delta = 25$, and $k_B$ is the Boltzmann constant. A grain was considered depinned if the system temperature $T \geq T_B$. If $T < T_B$, a local pinning field remained, increasing in strength with decreasing temperature. This was approximated by:

$$H_{pin} = H_0 \times (1 - T/T_B)^n \tag{5}$$

where $H_0 = 2 \times 10^7$ A·m$^{-1}$ and n = 1.

## Spike definition and network level drive interface

In this work, a spike (firing event) of the EB-MTJ neuron is defined as a thermally assisted MTJ state transition that switches between two resistance states ($R_{AP}$ to $R_P$). This resistance transition is treated as the device-level indicator that the neuron has crossed its effective switching threshold under Joule heating. To propagate this firing event through the network, the resistance transition is first converted into a standard electrical pulse by a compact sensing/comparator interface. A bias current $I_{bias}$ is applied to the neuron MTJ; upon switching, the sensing node exhibits a voltage step

$$\Delta V = I_{bias}\Delta R, \tag{6}$$

which is detected by a comparator and digitized as a fire pulse. This fire pulse then gates a driver that generates a calibrated voltage/current spike (with the required amplitude and width) delivered to the downstream synaptic stage for inter-layer communication and synapse activation. Although the EB-MTJ neuron does not intrinsically self-reset, the reset can be implemented using a reverse-polarity pulse through the same write path, with limited impact on scalability and efficiency under sparse spiking conditions. Developing EB-based neurons with intrinsic self-reset remains an important future direction.

## Convolutional spiking neural network simulations

CSNN simulations were carried out in SpikingJelly. All five convolutional layers used $3 \times 3$ kernels (stride 1, padding 1), followed by two fully connected layers. Synaptic plasticity in the output layer was implemented with a bi-exponential STDP rule whose time constants were set to $\tau_{pre} = 2$ and $\tau_{post} = 4$, values extracted from the temporal response of our EB-MTJ synapses. $\tau_{pre}$ and $\tau_{post}$ determine the decay of pre- and post-synaptic eligibility traces, thereby defining the effective temporal span over which paired spikes contribute to potentiation or depression. Leaky-integrate-fire neurons were calibrated to the device by fixing the resting potential at $V_{rest} = 300$ and the firing threshold at $V_{th} = 450$. The neuron constants $V_{rest}$ and $V_{th}$ map the EB-MTJ's thermally assisted barrier heights: cumulative sub-threshold SOT pulses raise the effective potential from 300 toward the 450 threshold, triggering a state switch that embodies the integrate-and-fire process. All neurons were modeled using parameters derived from the thermally assisted switching dynamics of EB-MTJs. Specifically, when presynaptic current pulses are applied, the local device temperature $T$ follows thermal accumulation with exponential memory, which we implement in discrete form as

$$T_{post} = \left(\frac{J_i^2}{K} + T_0\right)(1 - \alpha) + \alpha T_{pre}, \alpha = e^{-\tau/\tau_0}, \tag{7}$$

and during pulse cessation (no Joule heating),

$$T_{post} = T_0(1 - \alpha) + \alpha T_{pre}. \tag{8}$$

Here $J_i$ is the input current density, $K$ is an effective thermal factor capturing heat capacity and heat generation, $T_0$ is the ambient temperature, $\tau_0$ is the device thermal response time, and $\tau$ is the pulse interval. These equations explicitly realize integration (temperature accumulation under repeated pulses) and leak (exponential relaxation toward $T_0$), making $T(t)$ the physical state variable analogous to the membrane potential in an LIF neuron.

## Data availability

Source data are provided with this paper. Additional data are available from the corresponding author. Source data are provided with this paper.

## Code availability

The codes used in this paper are available from the corresponding author upon request.

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

## Acknowledgements

W.S.Z., K.W.S., and W.L.C. acknowledge the National Key Research and Development Program of China (Grant Nos. 2022YFB4400200, 2024YFA1408802) and the National Natural Science Foundation of China (Grant Nos. T2394473, T2394474). K.W.S. acknowledges the National Natural Science Foundation of China (Grant Nos. 62271026). W.L.C. acknowledges the National Natural Science Foundation of China (Grant Nos. 62401026). W.S.Z acknowledges the Beijing Outstanding Young Scientist Program. A.D. acknowledges the National Natural Science Foundation of China (Grant No. 623B2015). D.M.Z. acknowledges the Zhejiang Provincial Natural Science Foundation of China under Grant LZ24F040003.

## Author contributions

K.W.S., W.L.C., and W.S.Z. conceived the core idea of this study. Z.H.C. optimized and fabricated the samples. Z.H.C., W.L.C., and K.W.S. performed the experiment and measurement. Y.Z.S. and D.H.Z. performed the micro- and circuit-level simulations. K.W.S., W.L.C., Z.H.C., D.H.Z., A.D., Y.Z.S., Z.X.W., Y.Q.D., S.Y.L., K.H.C., H.Z., D.M.Z., H.X.L., and W.S.Z. conducted analysis. Z.H.C., D.H.Z., and Y.Z.S. prepared the manuscript. All authors contributed to the interpretation of the results and discussions.

## Competing interests

The authors declare no competing interests.
