## [Transparent Peer Review file · Nature Communications]

Nanoscale exchange-bias magnetic tunnel junctions enabled memristive synapse and leaky-integrate-fire neuron for neuromorphic computing

Corresponding Author: Dr Kewen Shi

Version 0:

Reviewer comments:

Reviewer #1

(Remarks to the Author)
See the attachment.

Reviewer #2

(Remarks to the Author)

In this work, the authors exchange bias CoFeB using IrMn. The devices are patterned into nanostructures < 100 nm in diameter, and round MTJs are found to have sharp switching with voltage through the Pt underlayer, while ellipse MTJs have graduate switching under the same voltage conditions. These are used to implement neurons and synapses, respectively.

It is appreciated that the authors measure these devices with TMR, which can be high enough SNR for application, contrasting with other works that use anomalous Hall effect. The micromagnetic and analytical analyses of the observed effect are also appreciated. The implementation of STDP and LIF function based on thermal heating + switching is an interesting approach.

While the results are intriguing, the following should be addressed before considering publication.

Major Comments:

- What is the role of the Fe heavy CoFeB composition?
- What is the role of the top IrMn?
- It is hard to see the crystallinity of the TEM in Fig. 1a and interpret the layers. It looks like the Pt is amorphous and the IrMn is sitting fairly crystalline on top of it. The bright area is the MgO, correct, it is hard to tell if it is crystalline as claimed. Please better justify the claim, "Cross-sectional TEM shows clear interfaces and crystallinity of the multilayer stack which are the essential characteristics for achieving high TMR ratio in MTJs."
- 70% TMR is fine, but not usually considered "high", especially for in-plane MTJs, as it is stated in the paper.
- Please clarify if the long axis of the ellipse is along y, the field annealing direction, and that the long axis is opposite the SOT current direction axis (x). It would be helpful to show in Fig 1a where the voltage V_{eff} is applied across the Pt heavy metal underneath, and have the figure actually show how the anisotropy is on the track, right now the angled view of the cartoon doesn't show that.
- Fig. 2a is overall very interesting that the many stable states can be achieved across the TMR range. But, MTJ based devices currently have the challenge of small on/off ratio relative to other technologies. This gives narrower room for each level across devices. Can you show Fig. 2a for more than one nominally identical device and see the spread in each R state? That will give a better idea how many states you actually can resolve, it is probably smaller than 25. E.g. see Fig. 7 of DOI 10.1109/IEDM45625.2022.10019564 .
- For Fig. 2f, usually a linear/symmetric pulse train is expected shown to see the linearity and symmetry over more than one

ramp up/down.

- The Fig 2c data was not clear, how is field immunity up to 2 T shown by the data? How does this compare to Fig. 1b where it is assumed a bigger field is used to set the device in the n- or n+ states?
- The pulses applied are very narrow (e.g. 0.4 ns) for the elliptical devices, can the authors comment on the shape of the pulse and if any reflections are expected? Was a ground/signal/ground setup needed? What is the behavior if the pulse is longer (~1-5 ns)?
- For the neural network application, can the authors add more details about how the device data is input into the model? Expand on the line, "All neurons were modeled using parameters derived from the thermally assisted switching dynamics of EB-MTJs."

Minor Comments:

- Please check grammar throughout, here are some examples:
 - o E.g. "on-chip co-integrated memristive synapse and leaky-integrate-fire (LIF) neuron designed.." should have "an" before
 - o Verb missing on line 47
 - o "we figures out" line 144
- If the nonuniform grain size of the IrMn is the source of multiple levels, then can the authors make any conclusions about the best way to grow the material for this application?

Reviewer #3

(Remarks to the Author)

-- The authors demonstrated various TMR values for the exchange bias MTJ device to show its multistate synaptic behavior. Since TMR values are highly sensitive to both temperature and voltage, a physics-based justification supported by equations is necessary to explain how the EB-based mechanism achieves the discrete 25 states shown in Figure-2. The provided Equations-1 and 2 are too generic in this context. In addition, the combined influence of temperature and voltage on TMR need be analyzed and discussed.

-- How about the impact of process variations? Since the margin between the states is very small, even small variations in material properties or fabrication steps can lead to significant mismatches in device behavior and performance.

-- In Figure-1(a), the authors present an SOT device, whereas in Figure-1(b), the switching mechanism corresponds to an STT device. However, SOT and STT are two fundamentally different technologies with distinct switching mechanisms. Please clarify this inconsistency. Additionally, why an in-plane MRAM structure considered when perpendicular MRAM is known to offer better switching efficiency (in terms of faster and low-energy switching)?

-- How about the retention capability of these devices? It is also not clear what cost is associated with the Joule heating effect in these EB-based MTJ devices, even though the authors claim energy-efficient, ultra-fast switching of 190 fJ/bit.

-- Authors have specified that they employed a hybrid back-propagation and STDP approach, where they claim back-propagation alleviates the vanishing plasticity problem, while STDP introduces biologically inspired local learning. However, the working principles of these two learning approaches are completely different, means difficult to integrate them. Secondly, it is not clear how the authors have dealt with the gradient mismatch problem in this case.

-- A general comments, authors are claiming high TMR, but >70% is not high enough, especially for the neuromorphic applications.

Version 1:

Reviewer comments:

Reviewer #1

(Remarks to the Author)

The authors have responded very well to my questions. Although I am not an expert in magnetic devices, from the perspective of a researcher working on neuromorphic devices, I find the implementation of second-order synaptic characteristics—enabling local learning rules to be realized intrinsically at the device level—to be particularly impressive. While the application of the neuron is somewhat limited due to the need for additional peripheral circuitry, the second-order characteristics are implemented in a stable manner, and the simulations based on these characteristics are conducted at a convincing and reasonable level.

Reviewer #2

(Remarks to the Author)

The authors have carefully addressed each point raised with modified or additional text, additional data, and modified or additional plots. I recommend acceptance.

Reviewer #3

(Remarks to the Author)

The authors have specified only the TMR values. However, the measured TMR depends on both the read voltage and temperature. Therefore, the corresponding bias voltage and temperature at which the TMR was measured need to be specified.

Response to referees' letter:

We sincerely thank the reviewers for their thorough evaluation of our manuscript and for providing insightful and constructive comments. We have carefully considered all suggestions and revised the manuscript accordingly. All changes are marked in blue, and additional analyses and discussions have been included in the Supplementary Information. Our point-by-point responses to the reviewers' comments are provided below:

Response To the Reviewer #1

This manuscript reports the development of memristive synapses and leaky-integrate-and-fire (LIF) neurons based on nanoscale exchange-bias magnetic tunnel junctions (EB-MTJs). The authors appear to utilize heat dynamics to realize synaptic learning behavior akin to so-called second-order synapses. The neuron functionality is also primarily based on thermal effects. While the demonstration of second-order synaptic behavior in EB-MTJ devices is intriguing, the mechanistic explanation remains insufficient, and for the neuron device it is unclear whether genuine neuronal operation has been achieved. Some specific questions are as follows:

Response: We thank the reviewer for the constructive and insightful comments. We appreciate the reviewer's careful assessment of this work and the important concerns raised regarding the underlying mechanisms and the neuronal functionality. We have carefully addressed all comments and revised the manuscript accordingly. The point-by-point responses are provided below.

Q1. The manuscript reports that elliptical EB-MTJ devices exhibit gradual, analog-like switching while circular devices show abrupt binary transitions. This distinction is crucial for assigning synaptic versus neuronal roles. However, the physical origin of this geometry dependent behavior is not sufficiently explained. Could the authors provide a more detailed discussion of how device shape (aspect ratio, demagnetizing fields, domain wall nucleation sites) influences the spin-orbit torque efficiency and switching characteristics? Is the observed difference reproducible across a wide range of device sizes and aspect ratios?

Response: We thank the reviewer for the constructive comments and suggestions. We apologize for not providing a sufficiently detailed physical description in the previous version. In response, we have carried out additional micromagnetic simulations and carefully compared them with the experimental results, which allows us to further clarify the physical origin. The distinct switching behaviors observed in elliptical and circular devices, namely progressive (analog-like) versus abrupt (binary-like) reversal, respectively, are not determined by the device geometry alone. The aspect ratio, device size, demagnetization fields, and domain walls can properly influence the device's properties.

Fig. R1 | Effect of aspect ratio on the number of intermediate states in elliptical devices.

First, we focused on the geometric parameters, and found that the aspect ratio plays the most critical role in enabling multistate behavior. As shown in Fig. R1, increasing the aspect ratio leads to a pronounced increase in the number of stable intermediate states, a trend that is consistently reproduced under different material conditions investigated in this study. The underlying physical origin of the aspect-ratio-dependent multistate behavior can be more clearly understood by considering the dynamic switching process. During the application of the writing current, the combination of the spin-orbit torque and Joule heating leads to a partial decoupling of antiferromagnetic grains from the ferromagnetic layer. As a result, the local exchange bias and the associated pinning are temporarily suppressed, allowing the corresponding regions of the ferromagnet to reverse more readily under the Oersted field. This process gives rise to intermediate multi-domain configurations in the ferromagnetic layer. After releasing the writing current, the stability of these intermediate configurations becomes strongly dependent on the device's aspect ratio. In elliptical devices with a higher aspect ratio, domain walls preferentially distribute along the short axis, resulting in a shorter domain-wall length and a lower total domain wall energy. These low-energy multi-domain states can remain stable for a sufficiently long time, enabling the previously decoupled antiferromagnetic grains to re-establish exchange coupling with the ferromagnet. This re-coupling process generates a newly oriented exchange-bias pinning field, which effectively locks the corresponding ferromagnetic domains into stable configurations. Consequently, a larger number of robust and distinguishable resistance states can be formed and retained over long timescales.

In addition to the aspect ratio, the lateral size of the device also influences whether the magnetization reversal proceeds in a binary-like or progressive manner as shown in Fig. R2. When the device size is sufficiently small (typically below ~ 50 nm), the ferromagnetic layer tends to remain in a single-domain state. In this regime, even under spin-orbit torque (SOT)-induced heating and partial suppression of exchange bias, the formation of stable multidomain configurations is energetically unfavorable, leading to predominantly abrupt, binary switching behavior. As the lateral size increases, the ferromagnetic layer can support localized or multi-domain reversal during the application of the writing current, and intermediate resistance

states appear in circular devices. However, owing to the circular geometry, the associated domain walls are typically long, resulting in a relatively high domain-wall energy. After the writing current is removed, these high-energy multidomain configurations are unstable and tend to collapse back to the initial state on a short timescale, often before the antiferromagnetic grains can fully re-establish exchange coupling with the ferromagnet. Consequently, only a limited number of intermediate states can be retained in large circular devices. In contrast, high-aspect-ratio elliptical devices accommodate domain walls with significantly shorter lengths, thereby reducing the domain-wall energy of the intermediate configurations. This allows the multidomain states to remain stable for a sufficiently long time after the end of the writing pulse, enabling the reformation of exchange bias and the establishment of robust pinning. As a result, while increasing the device size promotes the emergence of multidomain or localized reversal, the ability to realize a large number of stable and well-separated resistance states is primarily governed by whether the device geometry permits these configurations to be stabilized with a low domain-wall energy penalty, highlighting the dominant role of the aspect ratio.

Fig. R2 | Switching behavior of circular devices with different lateral sizes.

We further systematically examined other magnetic factors that may influence the switching behavior. Regarding the demagnetization field, we tuned the demagnetization energy term in micromagnetic simulations by varying the free-layer thickness and compared thicknesses of 1.8 nm and 2.0 nm (Fig. R3). For free-layer thicknesses close to the experimentally used value of 1.9 nm, variations in thickness have only a minor effect on the switching behavior. In this regime, thickness mainly modifies the overall energy barrier, while the switching pathway, the number of

accessible intermediate states, and their dependence on the aspect ratio and the device size remain essentially unchanged. This demonstrates that, around 1.9 nm, the free-layer thickness does not play a dominant role in determining whether the device exhibits progressive multistate behavior or abrupt binary switching.

Fig. R3 | Effect of demagnetization field on switching behavior.

Meanwhile, we also investigated the influence of domain-wall nucleation-site distribution on the switching characteristics (Fig. R4). By statistically comparing different initial nucleation positions at the device edges and in the interior, we find that the nucleation sites primarily affect the spatial unfolding of the reversal process and the sequence of local switching events, but do not alter the number of stable intermediate states nor their overall dependence on the aspect ratio in a statistical sense. In other words, the nucleation-site distribution determines where the reversal initiates, but not whether multistate behavior can be formed, and therefore does not constitute the fundamental origin of the observed multilevel resistance states. Moreover, as for the factor of SOT efficiency, the spin-orbit torque is mainly generated by the spin Hall effect in the bottom Pt layer. Since the material system and injection scheme are identical for different geometries, both the efficiency and direction of the spin-orbit torque remain essentially the same across devices, and thus are not responsible for the observed differences between progressive and abrupt switching behaviors.

Fig. R4 | Effect of nucleation-site distribution on switching behavior.

Finally, compared with the simulation results, we also conducted experimental tests on the different devices. Fig. R5(a–c) presents representative writing loops of circular EB-MTJs measured across a wide range of lateral sizes under 1-ns SOT pulses. The data consistently show that, as the device size increases, intermediate resistance states progressively emerge, indicating a reproducible transition from nearly coherent to multidomain or localized reversal during the writing process. Furthermore, when the device geometry is systematically modified to increase the aspect ratio, as shown in Fig. R5(d–i), the write windows associated with different resistance states become wider and more clearly separated, and a larger number of distinct multistate levels can be reliably accessed. These trends are observed reproducibly across devices with different sizes and aspect ratios, confirming that the geometry-dependent switching behavior is robust rather than device-specific. In agreement with the simulation analysis, the experimental results also demonstrate that the multistate capability of the present system is governed by a combination of the aspect ratio and lateral size. The aspect ratio is particularly effective in stabilizing multiple intermediate states, whereas the lateral size plays a crucial role in determining whether multidomain or nearly coherent switching occurs, especially when the device dimensions are reduced down to tens-of-nanometers regime. Based on this physical picture, we employ high-aspect-ratio elliptical devices to obtain rich and stable multistate behavior for synaptic units, while small circular devices are selected to realize nearly coherent binary switching for neuronal units, thereby establishing a clear and robust functional separation at the device level.

Fig. R5 | Influence of device size and aspect ratio on SOT-induced switching. **a–c** Switching loops of circular devices with varying diameters under a 1-ns SOT pulse. As the device size increases, a slight increase in the number of intermediate resistance states is observed during switching. **d–i** Switching traces of elliptical devices with different sizes and aspect ratios under a 1-ns SOT pulse. A pronounced increase in the number of intermediate resistance states is observed with increasing device size and aspect ratio.

This detailed discussion of the physical origin of the binary and multilevel switching characteristics has been added in *Section 3 of the Supplementary Information*. Correspondingly, the following clarification has been added to the revised manuscript at line 126:

“This distinction can be attributed to the reversal mode of the free layer: sufficiently small devices favor quasi-single-domain switching, while devices with larger lateral sizes and higher aspect ratios support non-uniform reversal involving relatively stable magnetic domain configurations, enabling multiple intermediate resistance states (see SI Section 3 for details). The formation and stability of these multi-state configurations are discussed in the following section.”

Q2. Throughout the manuscript, the authors rely on very short pulse durations (as low as 0.4 ns) and Δt timing to demonstrate both synaptic plasticity and neuronal dynamics. However, the actual pulse waveforms are not shown. Providing oscilloscope traces of the applied voltage/current pulses would significantly aid interpretation of the experiments, allowing readers to assess pulse fidelity, rise/fall times, and possible distortions at short Δt .

Response: We thank the reviewer for the question regarding pulse waveform fidelity, which helps improve the clarity and credibility of the manuscript. In this work, two types of programming signals are employed: single SOT pulses with pulse widths as short as 0.4 ns, and pulse trains consisting of up to ten consecutive pulses with variable inter-pulse intervals. To clarify this point, we present the experimentally measured waveforms of both signal types using a high-resolution oscilloscope.

Fig. R6 | Schematic of the pulse width and rise/fall times of a 0.4 ns pulse.

Fig. R6 shows a representative 0.4 ns SOT pulse generated by an arbitrary waveform generator, with a nominal amplitude of 1 V after amplification. The measured waveform demonstrates that the pulse maintains a well-defined shape at the sub-nanosecond timescale, with a rise time of approximately 157 ps, a fall time of approximately 160 ps, **both defined using the 10%–90% voltage criterion**, and an effective pulse width of about 407 ps. The measured pulse amplitude is ~ 1.007 V, in excellent agreement with the nominal post-amplification value, confirming the high fidelity and accuracy of the applied SOT pulses.

Fig. R7 | Schematic of pulse waveforms with different pulse widths.

Meanwhile, we also presented the waveform for single SOT pulses with different pulse widths. As shown in Fig. R7, pulses with widths of 0.4, 0.8, 2, and 5 ns all exhibit stable and well-defined waveforms. The corresponding rise/fall times are approximately 160, 162, 170, and 179 ps, respectively, indicating that the single-pulse signals used for device switching are robust over the investigated timescale range.

Fig. R8 | Schematic of a sequence of ten consecutive 0.4 ns pulses with different time intervals.

In addition, we displayed the fidelity of the pulse sequences employed to implement the leaky-integrate-and-fire (LIF) neuron functionality. Fig. R8 shows the measured waveforms of pulse trains consisting of ten consecutive 0.4 ns SOT pulses with different inter-pulse intervals. The pulse trains retain high signal fidelity across all tested conditions – even at an extremely short interval of 0.4 ns, the pulse amplitude and temporal profile remain comparable to those of an isolated single pulse, without observable distortion induced by pulse crowding. These results collectively confirmed the accuracy and reliability of the pulse waveforms used to realize the LIF neuron operation.

To clearly illustrate the coplanar waveguide compatibility of our devices and the shape of the sub-nanosecond SOT pulses used for switching, we have revised the main text at lines 117 and 278 of the revised manuscript. Specifically, the text at line 117 now reads:

“Moreover, the resistance state of the EB-MTJ can be modified by applying write voltage pulses through the bottom electrode, with pulse widths down to 0.4 ns as shown in Figs. 1c and 1d. The detailed pulse shapes and measurement procedures are described in the Methods and in Section 1 and 2 of the Supplementary Information (SI).”

And the text at line 278 now reads:

“..., the device was stimulated by a train of ten 0.4 ns SOT pulses at 20 ns intervals (exact pulse shapes are provided in Section 1 of the SI),...”

Both modifications are supported by additional discussion shown in *Section 1 of the Supplementary Information*.

Q3. In Fig. 3, the manuscript demonstrates LIF neuron characteristics, describing a “firing” event. In biological neurons, firing typically refers to the generation of an action potential or voltage spike. In the proposed EB-MTJ neuron, however, it appears that firing corresponds to a resistance state transition rather than a transient voltage output. Could the authors clarify how “firing” is defined in their device? Does this resistance change directly translate to a measurable voltage/current spike suitable for network-level signaling, or is additional circuitry required to emulate a biological-like spiking output?

Response: We apologize for the ambiguity in our previous main text. In the EB-MTJ neuron, “firing” is defined as a thermally assisted switching event of the neuron MTJ, i.e., a transition between two resistance states (e.g., high-resistance to low-resistance). This switching process represents the neuron crossing its effective threshold (reduced by Joule-heating), and thus serves as the physical correlate of an action-potential event in our device-level neuron model.

While the switching does not inherently produce a transient voltage spike identical to a biological action potential, we have designed a dedicated peripheral circuit to convert the MTJ resistance transition into a standard voltage/current pulse for network-level signaling. In our related work (accepted by IEEE DATE 2026)¹, we use a neuron array architecture consisting of an input voltage divider for aggregating weighted synaptic outputs, a core TI-MTJ LIF circuit, and a pulse monitoring unit. The voltage divider receives the summed synaptic output and applies the resulting gate voltage to transistors forming a current mirror (N1/N2). When synaptic input activates N1, the current mirror generates an identical write current I_{neu} through both the neuron MTJ and a reference MTJ placed under symmetric thermal conditions but configured to exhibit an opposite or non-switching response. Joule heating from I_{neu} raises the neuron-MTJ temperature, reducing its switching threshold. Once the threshold is exceeded, the neuron MTJ switches, representing our firing event, while the reference MTJ remains unchanged and provides a stable comparison point.

Fig. R9 | Schematic of a compact circuit to convert the resistance change to a current spike¹.

To generate a spike readout, the pulse monitoring unit continuously senses the neuron-MTJ resistance using a bias current source I_{bias} and a dynamic comparator. When the neuron MTJ switches, the voltage at the sensing node changes by

$$\Delta V = I_{bias} \Delta R,$$

which is captured by the comparator in real time and converted into a digital-level pulse (“Fire”). Therefore, the proposed neuron supports biological-like spiking outputs at the circuit level, where the MTJ state transition is the physical threshold-crossing event and the monitoring circuit produces a standard voltage/current pulse for downstream network communication.

To clarify how the resistance change after the device firing event is converted into an electrical pulse suitable for signal propagation in the network, we have added a brief description in the main text (line 284):

“Once the energy barrier drops below a critical threshold, a subsequent pulse triggers the “fire” event, which can be read out via appropriate sensing circuitry (see Methods for details).”

And provided a more detailed explanation in the *Methods* section (line 439):

“Device-Level Spike Definition and Network-Level Drive Interface

In this work, a spike (firing event) of the EB-MTJ neuron is defined as a thermally assisted MTJ state transition that switches between two resistance states (R_{AP} to R_P). This resistance transition is treated as the device-level indicator that the neuron has crossed its effective switching threshold under Joule heating. To propagate this firing event through the network, the resistance transition is first converted into a standard electrical pulse by a compact sensing/comparator interface. A bias current I_{bias} is applied to the neuron MTJ; upon switching, the sensing node exhibits a voltage step

$$\Delta V = I_{bias} \Delta R,$$

which is detected by a comparator and digitized as a “Fire” pulse. This “Fire” pulse then gates a driver that generates a calibrated voltage/current spike (with the required amplitude and width) delivered to the downstream synaptic stage for inter-layer communication and synapse activation. Although the EB-MTJ neuron does not intrinsically self-reset, the reset can be implemented using a reverse-polarity pulse through the same write path, with limited impact on scalability and efficiency under sparse spiking conditions. Developing EB-based neurons with intrinsic self-reset remains an important future direction.”

[1] Li, C.; Jiang, L.; Li, L.; Zhu, D.; Zhao, J.; Liu, H.; Kang, W.; Cai, W.; Zhang, H.; Zhao, W., “An Effective SNN Macro with Real-Time STDP and Dynamic LIF Model Based on Thermally Interplayed Spin-Orbit Torque MTJ,” Proc. IEEE Design, Automation & Test in Europe Conference (DATE), accepted for publication, 2026.

Q4. In biological neurons, firing is followed by an automatic return to the resting potential, providing intrinsic self-reset functionality. For the EB-MTJ neurons presented in this work, it is not clear whether the device spontaneously relaxes back to its initial state after a firing event, or whether external intervention is required. Do the EB-MTJ neurons exhibit genuine self-reset behavior, and if so, what is the underlying mechanism that distinguishes them from EB-MTJ synapses, which maintain non-volatile resistance states? If additional circuitry or stimuli are required for reset, how does this affect the scalability and efficiency of the proposed neuromorphic system?

Response: Thank you for the valuable comments. The EB-MTJ neuron in the present work does not exhibit intrinsic self-reset behavior. In our device, a “firing” event corresponds to an MTJ state transition. Because the MTJ state is non-volatile, the device does not spontaneously relax back to its pre-fire state after switching. Therefore, an external reset operation is required after each firing event. As Fig R10 shows, when a firing event is detected, we apply a reverse-polarity current pulse to deterministically switch the neuron MTJ back to its initial state. The reset pulse can be delivered through the same write path used for firing, with controlled amplitude and duration to ensure reliable return. Functionally, the neuron operates as an integrate-and-switch element, and reset is implemented by a reverse stimulus following each firing.

Fig. R10 | neuron device operation diagram with integration, firing and leaky process

We acknowledge that the need for an explicit reset introduces additional circuitry/stimulus compared with an ideal self-reset neuron. However, the overhead can be kept modest for the reason that neuron switching is intrinsically fast within nanoseconds and the reset operation is triggered only upon firing events, remaining the overall consumption at the scale of hundreds of femtojoules. Furthermore, our system operates in a sparse spiking regime, resulting in a low reset duty cycle. From an architectural perspective, reset can be implemented using shared reset drivers/lines per neuron block (row/column or subarray) and a simple control handshake with the spike-monitoring circuit. In addition, the EB-MTJ neuron itself has a compact footprint (device dimensions in the tens-of-nanometers range), which is favorable for dense integration and large-scale neuromorphic arrays.

We agree with the reviewer that intrinsic self-reset is a desirable feature. Indeed, several spintronic neuron concepts with self-reset or relaxation dynamics have been proposed/demonstrated in the literature (e.g., designs based on intrinsic relaxation/oscillation mechanisms). However, these schemes are generally not directly compatible with our present EB-MTJ stack and operation mode, because the exchange-biased EB-MTJ is engineered for robust non-volatile states and thermally assisted threshold switching. Enabling true self-reset while preserving the EB-MTJ benefits (thermal integration, stability, multi-state compatibility) is an important direction for future optimization and innovation.

To explain the reset mechanism of the neuron device and to confirm that the adopted global reset scheme preserves the scalability and energy efficiency of the proposed neuromorphic architecture, we have added the following clarification in the revised manuscript at line 450:

“Although the EB-MTJ neuron does not intrinsically self-reset, the reset can be implemented using a reverse-polarity pulse through the same write path, with limited impact on scalability and efficiency under sparse spiking conditions. Developing EB-based neurons with intrinsic self-reset remains an important future direction.”

Q5. For a device to function as a practical spiking neuron, its firing event must generate an output signal strong enough to drive subsequent synapses in a network. Can the proposed neuron device directly provide the necessary voltage or current pulse to activate downstream EB-MTJ synapses?

Response: In our current device concept, the EB-MTJ “firing” event is defined as an MTJ resistance-state transition and does not by itself generate a sufficiently strong transient voltage/current pulse to directly program or activate downstream EB-MTJ synapses. Therefore, similar to many MTJ-based neuromorphic implementations, external peripheral circuitry is required to translate the resistance transition into a network-level spike signal.

Concretely, a compact pulse monitoring unit detects the resistance change and produces a digital-level “Fire” pulse. This pulse can then enable a dedicated driver to generate a calibrated voltage/current spike (with appropriate amplitude and width) that is delivered to the synaptic array of the subsequent layer. We will revise the manuscript to explicitly state this separation of functions, device-level firing as a resistance transition, and circuit-level spike generation for inter-layer communication and synapse activation, and we will discuss the scalability implications (shared drivers per neuron block and event-driven activation so the overhead scales with spike activity rather than with array size).

Fig. R11 | Schematic of a compact circuit to convert the resistance change to a current spike.

To clarify how the device provides a propagating electrical pulse to the downstream network after the firing event, we have added the following description in the revised manuscript (line 284):

“Once the energy barrier drops below a critical threshold, a subsequent pulse triggers the “fire” event, which can be read out via appropriate sensing circuitry (see Methods for details).”

Q6. The simulation section lacks sufficient detail to assess how faithfully device-level dynamics were incorporated into the network model. Since the EB-MTJ operation relies heavily on

heat-assisted switching, one would expect not only STDP but also other forms of learning dynamics (e.g., BCM-type plasticity) to emerge from the thermal processes. Could the authors clarify whether their simulations include heat dynamics beyond simple STDP fitting? Was the possibility of BCM-like or other second-order learning rules considered, and if so, how might they affect network-level behavior?

Response: We thank the reviewer for this helpful comment and appreciate the opportunity to clarify how experimentally measured EB-MTJ device characteristics are incorporated into the network simulations. In our simulation framework, the network model is informed by multiple experimentally observed aspects of the devices. In particular, the multilevel resistance states define the synaptic weights, the measured switching statistics and relaxation characteristics inform the synaptic update dynamics, and the heat-assisted switching and relaxation behavior of EB-MTJ neuron devices governs the neuronal integration and firing processes. Following the reviewer's suggestion, we additionally explored higher-order learning rules enabled by the same thermally accumulated device dynamics, including triplet-based STDP and the associated BCM-like plasticity. While these higher-order rules are not required for the main results, we find that the BCM-like learning rule leads to faster convergence and more rapid stabilization during network training. Below, we describe in more detail how these experimentally derived device characteristics are incorporated into the synaptic and neuronal components of the model.

Quantization of EB-MTJ Device States for Weight Storage

In our model, the weights are stored in the synaptic connections as quantized values derived from the 25 resistance states of EB-MTJs. These resistance states are mapped to the convolutional and fully connected layers of the network. This mapping allows us to use the multi-level resistance states of the EB-MTJ devices, thereby enhancing the precision of weight storage. We compared the accuracy of the network with different quantization levels: 5 levels, 15 levels, and 25 levels. The accuracy curve clearly demonstrates the advantage of using 25 levels, as it outperforms the 5-level and 15-level quantization schemes, showcasing the potential of EB-MTJ in efficiently storing more precise weight values.

Fig. R12 | Comparison between real weights and quantized weights in our neuron network. **a,b** The real weights and quantized weights in our neuron network. The input index and output index denote neuron indices in the previous and next layer, respectively; each pixel is the weight $W_{j,i}$ connecting input neuron i to output neuron j . **c** The difference ($W_{quant} - W_{real}$) of the real weights and quantized weights.

During training, we introduce quantization-aware training (QAT), a gradual quantization process controlled by a parameter α , which interpolates between the real-valued weights and the quantized EB-MTJ states¹. The effective weight w_{eff} at each training step is given by the following equation:

$$w_{eff} = (1 - \alpha)w_{real} + \alpha Quant(w_{real})$$

where w_{real} is the real-valued weight computed from the training process, and $Quant(w_{real})$ is the quantized weight derived from the EB-MTJ states. The parameter α smoothly increases from 0 to 1 during training, with $\alpha = 0$ corresponding to no quantization (fully real-valued weights) and $\alpha = 1$ corresponding to complete quantization (fully discrete weights). This smooth transition ensures that the network stabilizes early in training using real-valued weights and gradually adapts to the quantized weights as training progresses.

Fig. R13 | a The test accuracy of different numbers of state. **b** QAT controlled parameter α changed from 0 to 1 within 20 epoch.

Neuron and synapse implementation in our CSNN

Thermally assisted integration–leak dynamics in EB-MTJ neurons.

Each EB-MTJ neuron emulates membrane potential integration and leakage through its heat-assisted switching dynamics. When input current pulses I from pre-neurons are applied, the local device temperature T increases according to the thermal accumulation equation²:

$$T_{post} = \left(\frac{J_i^2 \alpha}{K} + T_0\right)(1 - \alpha) + T_{pre} \alpha,$$

where $\alpha = e^{-\tau/\tau_0}$, J_i is the input current density, K represents the combined specific heat and heat generation factor, T_0 is the ambient temperature, and τ_0 is the device-specific thermal response time. During current cessation, the device naturally cools down following

$$T_{post} = T_0(1 - \alpha) + T_{pre}\alpha,$$

which directly emulates the leakage of membrane potential in biological neurons. As a result, by leveraging the thermal effect, we achieve the integrity and leaky process of neuron. What's more, as the spikes are successively injected, the temperature of our device reached the threshold (T_{th}), leading to the switch of EB-MTJ, representing the fire process. Importantly, the temperature accumulation and threshold switching behavior used in the neuron model is mapped from measured device characteristics rather than assumed phenomenologically. As evidenced in Fig. R14, changing the pulse width modulates the effective heating and thus the switching condition: the extracted switching temperature/thermal background and the pulse-width-dependent critical switching amplitude V_{eff} together quantify how thermal activation governs the transition. In the simulations, we therefore extract τ_0 from the measured thermal relaxation behavior (consistent with the $\ln(\tau/\tau_0)$ scaling in Fig. R14a³) and use the experimentally determined switching threshold statistics to set the neuron's T_{th} (equivalent electrical threshold under a given pulse width).

Fig. R14 | **a** The estimated temperature during switching of devices³. **b** Critical switching V_{eff} of the device as a function of pulse width.

When mapped to synaptic learning, this thermal memory naturally produces exponentially decaying eligibility traces: a pre- (or post-) spike perturbs the device temperature (and hence the switching barrier) and its influence decays approximately as $e^{-\Delta t/\tau_0}$. Therefore, the commonly used exponential STDP kernel,

$$\Delta w(\Delta t) \propto \exp(-|\Delta t|/\tau_{\pm}),$$

is not an ad hoc assumption in our work, but an effective description consistent with the EB-MTJ's exponential thermal relaxation and the measured timing dependence of weight changes.

In other words, thermal dynamics provide the physical origin of the exponential traces underlying STDP.

Moreover, because temperature accumulation introduces state/history dependence beyond a single spike pair (e.g., multiple closely spaced spikes elevate the baseline temperature), the same thermal-trace mechanism can support higher-order plasticity. This motivates our extension from pair-based STDP to triplet-STDP, where a third spike interacts with residual thermal traces. Under standard rate averaging, the triplet rule reduces to a BCM-like meta-plasticity law with a firing-rate threshold θ . We thus carried out the BCM-like second-order effects as emerging naturally from thermal memory, rather than being introduced artificially.

Pair-based STDP

For the STDP training part, we employ the standard exponential pair rule:

$$\Delta w(\Delta t) = \begin{cases} A_+ \exp(-\Delta t/\tau_+) & \Delta t > 0, \\ -A_- \exp(+\Delta t/\tau_-) & \Delta t < 0, \end{cases}$$

where $\Delta t = t_{\text{post}} - t_{\text{pre}}$. The parameters A_{\pm} and τ_{\pm} are obtained by fitting experimentally measured weight-change (or resistance-change) curves under paired pre/post stimulation protocols. In our device fitting, we obtained $\tau_+ = 3.81\text{ns}$, $\tau_- = 2.70\text{ns}$, $A_2 = 0.787$, $A_2^+ = 0.789$ and incorporated them into our STDP training process.

Fig. R15 | STDP rule of long-term plasticity demonstrated for paired spikes

To simplify our training process and avoid the need to record all firing times of pre and post neurons, we use the trace method to implement STDP in our neuron network as depicted in Section 16 of the SI. The trace-based formulation of STDP:

$$tr_{pre}[i](t) = tr_{pre}[i](t-1)e^{-\frac{1}{\tau_{pre}}} + s_i(t)$$

$$tr_{post}[j](t) = tr_{post}[j](t-1)e^{-\frac{1}{\tau_{pre}}} + s_j(t)$$

$$\Delta W_{ij}(t) = F_{pot}(W_{ij}(t)) tr_{pre}[i](t) s_j(t) - F_{dep}(W_{ij}(t)) tr_{post}[j](t) s_i(t)$$

where $s_{ij}(t) \in \{0,1\}$ are spike indicators and $F_{\text{pot/dep}}$ are the bounded update functions. The chosen decay constants $\tau_{\text{pre}} = 2$ and $\tau_{\text{post}} = 4$ match the intrinsic relaxation of our EB-MTJ synapses τ_+ and τ_- as shown in Fig R15.

Triplet-STDP (considered to capture second-order effects)

To examine whether the thermally assisted SOT switching mechanism could produce higher-order learning dynamics, we also fitted a triplet-STDP model in which the weight update depends not only on spike pairs but also on triplets through additional state traces (capturing the “history dependence” introduced by thermal accumulation). Using the standard trace-based formulation, the update at a postsynaptic spike and a presynaptic spike can be written as⁴

$$\Delta W = \begin{cases} \Delta W^+ = e^{\frac{-\Delta t}{\tau_+}} (A_2^+ + A_3^+ e^{\frac{-\Delta t_1}{\tau_y}}) \\ \Delta W^- = -e^{\frac{\Delta t}{\tau_-}} (A_2^- + A_3^- e^{\frac{-\Delta t_2}{\tau_x}}) \end{cases}$$

where ΔW denotes the weight change of an EB-MTJ synapse induced by spike interactions. $\Delta W = \Delta W^+$ evaluated at $t = t_{\text{post}}$ corresponds to the post–pre–post triplet case (LTP), whereas $\Delta W = \Delta W^-$ evaluated at $t = t_{\text{pre}}$ corresponds to the pre–post–pre triplet case (LTD). $\Delta t = t_{\text{post}} - t_{\text{pre}}$ is the time interval between the presynaptic spike and the postsynaptic spike. t'_{pre} and t'_{post} denote the time instants of the previous presynaptic spike and the previous postsynaptic spike in the pre–post–pre or post–pre–post triplet sequence, respectively. Accordingly, $\Delta t_1 = t'_{\text{post}} - t_{\text{post}}$ is the interval between two consecutive postsynaptic spikes, and $\Delta t_2 = t'_{\text{pre}} - t_{\text{pre}}$ is the interval between two consecutive presynaptic spikes.

Parameters A_2^+ , A_2^- , A_3^+ , and A_3^- are the amplitude coefficients of pair-based and triplet-based potentiation/depression terms, respectively, while τ_+ and τ_- are the characteristic time constants for the pair timing dependence. τ_x and τ_y are the time constants governing the decay of the additional triplet “trace” terms associated with presynaptic and postsynaptic history, respectively⁴.

Fig. R16 | Triplet STDP fitting with the parameter τ_x , τ_y , A_3^+ and A_3^-

Triplet → BCM (minimal triplet rule and derived BCM threshold)

Fig. R17 | Typical triplet-pulse diagram with ‘pre–post–pre’ and ‘post–pre–post’ sequences

The triplet-STDP learning landscape can be used to generalize the BCM learning rule. The mathematical model of BCM learning can be expressed as^{4,5}

$$\frac{dW}{dt} = \rho_{pre} \phi(\rho_{post}, \theta),$$

where ρ_{pre} and ρ_{post} are the frequencies of presynaptic and postsynaptic spikes, respectively, and $\phi(\rho_{post}, \theta)$ is a scalar function of ρ_{post} with a threshold frequency θ . Here, we define $\rho_{pre} = 1/\Delta t_1$ and $\rho_{post} = 1/\Delta t_2$. Synaptic depression ($\Delta W < 0$) occurs if $\phi(\rho_{post} < \theta, \theta) < 0$, whereas synaptic potentiation ($\Delta W > 0$) occurs if $\phi(\rho_{post} > \theta, \theta) > 0$; no synaptic change occurs at the modification threshold where $\phi(\rho_{post}, \theta) = 0$.

By relating triplet-STDP to the BCM rule, Eq. can be further expressed as^{5,6}

$$\frac{dW}{dt} = -A_2^- \tau_- \rho_{pre} \rho_{post} - A_3^- \tau_- \tau_x \rho_{pre}^2 \rho_{post} + A_2^+ \tau_+ \rho_{pre} \rho_{post} + A_3^+ \tau_+ \tau_y \rho_{pre} \rho_{post}^2,$$

where A_2^\pm and A_3^\pm are the fitted amplitudes of the pair-based and triplet-based potentiation/depression components, respectively, and τ_+ , τ_- , τ_x , and τ_y are the corresponding time constants extracted from device-fitted triplet-STDP curves.

For the minimal triplet model (used to obtain an explicit BCM-like threshold), we set A_2^+ and A_3^- to 0 and the dominant LTD and LTP contributions reduce Eq. to a BCM-form rate dependence

$$\frac{dW}{dt} = \rho_{pre} \rho_{post} (-A_2^- \tau_- + A_3^+ \tau_+ \tau_y \rho_{post}) \propto \rho_{pre} \rho_{post} (\rho_{post} - \theta),$$

with the corresponding modification threshold

$$\theta = \frac{A_2^- \tau_-}{A_3^+ \tau_+ \tau_y}.$$

Using our experimentally fitted parameters, we obtain

$$\theta \approx \frac{2.7 \times 0.787 \text{ ns}}{322 \times 0.789 \text{ ns} \times 13 \text{ ns}} \approx 6.43 \times 10^5 \text{ Hz} \approx 0.643 \text{ MHz}.$$

This explicitly indicates a BCM-like transition: at low ρ_{post} the net update is depressive, whereas at sufficiently high ρ_{post} the triplet term dominates and yields potentiation, with the crossover set by θ . Final parameters extracted by the fitting curve can be concluded in the table 1

Parameters	A_2^+	A_2^-	τ^+ (ns)	τ^- (ns)	A_3^+	A_3^-	τ_x (ns)	τ_y (ns)
------------	---------	---------	---------------	---------------	---------	---------	---------------	---------------

Experimental data	3.8	2.7	0.790	0.787	322	62	4.6	13
Minimal triplet model	0	2.7	0.789	0.787	322	0	4.6	13

Table 1 Parameters extracted from paired-STDP and triplet-STDP, optimized using the minimal triplet model.

Fig. R18 | Triplet-STDP-based BCM learning rules

Network-level implementation of BCM learning

For the network implementation, we try to replace STDP with BCM training method to get the effect of BCM learning rules. Concretely, for the synaptic layers trained by local plasticity (the last fully connected layer), we record the pre- and postsynaptic spike trains over a time window of length T during each training step. From these spike trains we compute the empirical firing rates

$$\rho_{pre} = \frac{1}{TB} \sum_{t,b} s_{pre}(t,b), \rho_{post} = \frac{1}{TB} \sum_{t,b} s_{post}(t,b),$$

where b indexes batch samples. In the fully connected layer this yields one scalar $\rho_{pre}(j)$ per input neuron and $\rho_{post}(i)$ per output neuron. On top of these rates, we implement the minimal BCM form implied by Eq.:

$$\Delta W_{ij} = \eta \rho_{pre,j} \rho_{post,i} (\rho_{post,i} - \theta_i) g(W_{ij}),$$

where η is an overall learning rate and $g(W) = \text{clip}(W, -1, 1)$ plays the same role as $F_{pot/dep}$ in the original pair/triplet rule (it prevents weights from drifting outside the experimentally relevant range). The sliding modification threshold θ_i for each postsynaptic neuron is updated as an exponential moving average of the instantaneous postsynaptic activity,

$$\theta_i \leftarrow \theta_i - \frac{\theta_i}{\tau_\theta} + \frac{\rho_{post,i}^p}{\tau_\theta},$$

with $p = 2$, as in standard BCM, and τ_θ chosen of the same order as the slow triplet time constant τ_y . To ensure consistency with the device-level triplet-STDP fits, we initialize the

effective BCM hyper-parameters from the experimentally extracted amplitudes and time constants. Comparing the rate-averaged expression

$$\frac{dW}{dt} = \rho_{pre} [(-A_2^- \tau_- + A_3^+ \tau_+ \tau_y \rho_{post}) \rho_{post}] \approx \eta \rho_{pre} \rho_{post} (\rho_{post} - \theta),$$

gives $\eta \propto A_3^+ \tau_+ \tau_y$ and $\theta \propto (A_2^- \tau_-) / (A_3^+ \tau_+ \tau_y)$, as already stated. In the code we use these relations to set the initial scale of η and the initial value of θ , and we take $\tau_\theta \sim \tau_y$. Small dimensionless prefactors are then tuned within a narrow range to account for the discrete-time nature of training, the finite simulation window T , and numerical stability of the optimizer. We have verified that these rescalings do not change the qualitative BCM behavior: for low postsynaptic firing rates the net update is depressive, while above the fitted threshold θ the triplet term dominates and produces potentiation, as shown in Fig. R18.

Finally, we note that in the quantization-aware training (QAT) regime used for EB-MTJ synapses, the same BCM rule is applied to the underlying continuous weights W_{real} , while the forward pass uses an effective weight

$$W_{eff} = (1 - \alpha) W_{real} + \alpha W_q,$$

where W_q is the 25-state device-quantized weight and $\alpha \in [0,1]$ is a mixing parameter. As $\alpha \rightarrow 1$, the BCM learning rate is smoothly reduced and eventually frozen, so that the learned weights remain consistent with the experimentally calibrated discrete conductance states. In this way, the network-level plasticity rule used in our CSNN simulations is a direct rate-based implementation of the minimal triplet to BCM mapping derived above, with parameters anchored to the device-measured pair/triplet STDP curves.

Finally, Fig. R19 compares the test accuracy of networks trained with the STDP and BCM rules. The BCM-trained network shows a similar learning dynamics and converges to a test accuracy only $\sim 2\%$ lower than the STDP baseline. In this work the BCM configuration is used mainly to verify the BCM-type plasticity of our neuron-synapse model, and its hyper-parameters were not extensively optimized. With further tuning, the BCM accuracy can be reduced to a negligible gap or even exceed that of the STDP setting.

Fig. R19 | The comparison of test accuracy of STDP and BCM.

To provide a detailed explanation of how our device data are input into the neural network model, we have added the following description at line 313 of the revised manuscript:

“..., with the mapping of device-level parameters to the neuron model described in the Methods, and the detailed simulation procedures described in Section 11 of the SI.”

And provided a more detailed explanation in the Methods section (line 467):

“All neurons were modeled using parameters derived from the thermally assisted switching dynamics of EB-MTJs. Specifically, when presynaptic current pulses are applied, the local device temperature T follows thermal accumulation with exponential memory, which we implement in discrete form as

$$T_{post} = \left(\frac{J_i^2}{K} + T_0\right)(1 - \alpha) + \alpha T_{pre}, \alpha = e^{-\tau/\tau_0},$$

and during pulse cessation (no Joule heating),

$$T_{post} = T_0(1 - \alpha) + \alpha T_{pre}.$$

Here J_i is the input current density, K is an effective thermal factor capturing heat capacity and heat generation, T_0 is the ambient temperature, τ_0 is the device thermal response time, and τ is the pulse interval. These equations explicitly realize integration (temperature accumulation under repeated pulses) and leak (exponential relaxation toward T_0), making $T(t)$ the physical state variable analogous to the membrane potential in an LIF neuron.”

To demonstrate that our writing mechanism can not only implement standard STDP functionality but also support other plasticity rules such as BCM, we have added the following description at line 239 of the revised manuscript.

“Beyond standard STDP, the thermal-assisted writing mechanism can also implement other plasticity rules, such as BCM, which relies on the correlation between the average activity of the pre- and post-synaptic neurons. The underlying principle and experimental implementation of BCM using EB-MTJs are described in Section 10 of the SI.”

The detailed modeling procedures for mapping device-level parameters to the neural network, as well as the underlying principle and experimental implementation of the BCM rule based on EB-MTJs, are provided in *Sections 11 and 10 of the Supporting Information*, respectively.

- [1] Krestinskaya, L. Zhang and K. N. Salama, "Towards Efficient RRAM-based Quantized Neural Networks Hardware: State-of-the-art and Open Issues," *2022 IEEE 22nd International Conference on Nanotechnology (NANO)*, Palma de Mallorca, Spain, 2022, pp. 465-468, doi: 10.1109/NANO54668.2022.9928590.
- [2] D. Zhu *et al.*, "Thermally Driven Leaky-Integrate-and-Fire Spintronic Neurons With Stray-Field-Enabled Self-Reset for Neuromorphic Computing," in *IEEE Electron Device Letters*, vol. 46, no. 8, pp. 1437-1440, Aug. 2025, doi: 10.1109/LED.2025.3580276.

- [3] Cai, W. *et al.* Anatomy of Thermally Interplayed Spin-Orbit Torque Driven Antiferromagnetic Switching. Preprint at <https://doi.org/10.48550/arXiv.2410.13202> (2024).
- [4] Wang, Z., Zeng, T., Ren, Y. *et al.* Toward a generalized Bienenstock-Cooper-Munro rule for spatiotemporal learning via triplet-STDP in memristive devices. *Nat Commun* 11, 1510 (2020). <https://doi.org/10.1038/s41467-020-15158-3>
- [5] Nie, Fang, *et al.* "An Adaptive Solid-State Synapse with Bi-Directional Relaxation for Multimodal Recognition and Spatio-Temporal Learning." *Advanced Materials* 37.17 (2025): 2412006.
- [6] R. A. John, A. Milozzi, S. Tsarev, R. Brönnimann, S. C. Boehme, E. Wu, I. Shorubalko, M. V. Kovalenko, D. Ielmini, *Sci. Adv.* 2022, 8, eade0072.

Response To the Reviewer #2

In this work, the authors exchange bias CoFeB using IrMn. The devices are patterned into nanostructures < 100 nm in diameter, and round MTJs are found to have sharp switching with voltage through the Pt underlayer, while ellipse MTJs have graduate switching under the same voltage conditions. These are used to implement neurons and synapses, respectively. It is appreciated that the authors measure these devices with TMR, which can be high enough SNR for application, contrasting with other works that use anomalous Hall effect. The micromagnetic and analytical analyses of the observed effect are also appreciated. The implementation of STDP and LIF function based on thermal heating + switching is an interesting approach. While the results are intriguing, the following should be addressed before considering publication.

Response: We sincerely thank the reviewer for the positive and constructive evaluation of our work. We appreciate the reviewer's recognition of our device design, the use of TMR readout, and the micromagnetic and analytical analyses. We also thank the reviewer for noting the interest of our STDP and LIF implementations based on thermally assisted switching. We have carefully addressed all the concerns raised by the reviewer and revised the manuscript accordingly. Detailed point-by-point responses to each comment are provided below.

Q1. What is the role of the Fe heavy CoFeB composition?

Response: We appreciate the reviewer's insightful question regarding the role of the Fe heavy CoFeB composition. In MgO-based magnetic tunnel junctions, the exceptionally high tunneling magnetoresistance (TMR) fundamentally originates from coherent spin-dependent tunneling dominated by the $\Delta 1$ symmetry states¹⁻³. In crystalline CoFeB/MgO/CoFeB stacks, the MgO barrier selectively transmits $\Delta 1$ Bloch states, which are highly spin-polarized at the Fermi level in

bcc ferromagnets, leading to a pronounced conductance contrast between the parallel and antiparallel magnetic configurations.

An Fe heavy CoFeB composition is particularly favorable for realizing and stabilizing this $\Delta 1$ -dominated coherent tunneling mechanism. Previous studies by S. Ikeda et al^{4,5} have demonstrated that in $\text{Co}_x\text{Fe}_{80-x}\text{B}_{20}/\text{MgO}/\text{Co}_x\text{Fe}_{80-x}\text{B}_{20}$ junctions with identical layer thicknesses, the Fe-rich samples ($x \approx 20\text{--}25\%$) exhibit ultrahigh TMR ratios approaching 470–500 % after post-annealing, significantly outperforming compositions with lower Fe content. Moreover, even after high-temperature annealing at 600 °C, the TMR remains above 300 %, highlighting the excellent thermal robustness of Fe-rich CoFeB/MgO systems. This superior TMR performance and thermal stability are closely associated with the annealing-induced crystallization behavior at the CoFeB/MgO interface⁶. Upon annealing, MgO promotes the formation of a highly ordered bcc(001) texture in adjacent Fe-rich CoFeB layers, with $\text{Co}_{20}\text{Fe}_{60}\text{B}_{20}$ beginning to crystallize around ~ 325 °C and achieving near-complete bcc(001) ordering before ~ 475 °C. The high Fe concentration facilitates better lattice matching and reduces interfacial disorder and defect density, which is crucial for preserving phase coherence of the $\Delta 1$ tunneling channels.

The boron component plays a complementary but equally important role^{7,8}. During deposition, B suppresses premature polycrystalline growth, enabling the formation of an initially amorphous CoFeB layer with a smooth interface. During annealing, B diffuses out of the CoFeB layer into adjacent layers (e.g., Ta-based buffers or caps), leaving behind a high-quality, well-ordered bcc CoFe lattice. However, the B content must be carefully optimized, as excessive B would hinder crystallization and degrade the coherent tunneling condition.

In addition to its favorable tunneling properties, the Fe-rich CoFeB composition also increases the saturation magnetization and spin polarization, which benefits current-induced magnetization switching. This enhanced magnetic moment facilitates efficient SOT- or Oersted-field-driven switching of the free layer and improves the overall electrical and magnetic controllability of the device, making Fe-rich CoFeB an optimal choice for both high-performance and thermally robust MTJs.

To clearly clarify the role of Fe-rich CoFeB in the CoFeB/MgO tunnel junction, we added the text at line 94 of the manuscript:

“Here, we employ an Fe-rich CoFeB alloy as the ferromagnetic layer, which is known to provide high tunneling magnetoresistance (TMR) and excellent thermal stability in MgO-based MTJs”

[1] Butler, W. H., Zhang, X.-G., Schulthess, T. C. & MacLaren, J. M. Spin-dependent tunneling conductance of Fe|MgO|Fe sandwiches. Phys. Rev. B 63, 054416 (2001).

- [2] Mathon, J. & Umerski, A. Theory of tunneling magnetoresistance of an epitaxial Fe/MgO/Fe(001) junction. *Phys. Rev. B* 63, 220403 (2001).
- [3] Butler, W. H. Tunneling magnetoresistance from a symmetry filtering effect. *Sci. Technol. Adv. Mater.* 9, 014106 (2008).
- [4] Ikeda, S. et al. A perpendicular-anisotropy CoFeB–MgO magnetic tunnel junction. *Nature Mater* 9, 721–724 (2010).
- [5] Ikeda, S. et al. Tunnel magnetoresistance of 604% at 300K by suppression of Ta diffusion in CoFeBMgO/CoFeB pseudo-spin-valves annealed at high temperature. *Appl. Phys. Lett.* 93, 082508 (2008).
- [6] Yuasa, S., Suzuki, Y., Katayama, T. & Ando, K. Characterization of growth and crystallization processes in CoFeBMgO/CoFeB magnetic tunnel junction structure by reflective high-energy electron diffraction. *Appl. Phys. Lett.* 87, 242503 (2005).
- [7] Yuasa, S. & Djayaprawira, D. D. Giant tunnel magnetoresistance in magnetic tunnel junctions with a crystalline MgO(0 0 1) barrier. *J. Phys. D: Appl. Phys.* 40, R337 (2007).
- [8] Djayaprawira, D. D. et al. 230% room-temperature magnetoresistance in CoFeBMgO/CoFeB magnetic tunnel junctions. *Appl. Phys. Lett.* 86, 092502 (2005).

Q2. What is the role of the top IrMn?

Response: We thank the reviewer for the question regarding the top IrMn layer, this helps clarify the design and functionality of our device. Owing to its high blocking temperature and the ability to provide a large exchange bias when interfaced with ferromagnetic layers, IrMn has become a standard antiferromagnetic pinning material and is widely used in GMR read heads and in-plane anisotropy MRAMs¹⁻³.

Fig. R20 | Schematic model of the film structure.

As illustrated in Fig. R20, the layer stack of our device incorporates a synthetic antiferromagnet (SAF) composed of CoFe/Ru/CoFeB adjacent to the top IrMn layer. The IrMn establishes a strong exchange bias with the upper CoFe layer, rigidly fixing its magnetization

direction. This pinned CoFe layer, in turn, stabilizes the reference-layer magnetization through the Ru-mediated antiferromagnetic RKKY coupling within the SAF, thereby ensuring robust reference-layer stability against SOT-induced perturbations. This design is particularly important because the device is written using SOT, which acts simultaneously on both the free and reference layers. Simply placing an antiferromagnetic layer above the reference layer is not sufficient, as the exchange bias experienced by the reference layer may be weak or comparable to that on the free layer, and stray fields from the reference layer could still influence free-layer switching^{4,5}. To address this, synthetic SAF structures, such as the CoFe/Ru/CoFeB trilayer used here, are commonly employed. The Ru-mediated antiferromagnetic RKKY coupling provides much stronger coupling (typically several thousand Oersted field), while the near-zero net moment of the SAF minimizes its influence on the free layer^{6,7}. Furthermore, because the top CoFe layer does not contribute to the TMR, a harder ferromagnetic material can be used to enhance the exchange bias that stabilizes the CoFe and reference-layer magnetization, thereby improving overall device stability.

To provide a clearer explanation of the role of top IrMn in the film stack, we have revised the text at line 96 of the manuscript:

“The top IrMn antiferromagnet (AFM) layer, together with the RKKY-coupled SAF, stabilizes the magnetization of the reference layer. In addition, the bottom IrMn layer stabilizes the magnetization of CoFeB via exchange bias, ...”

[1] Szunyogh, L., Lazarovits, B., Udvardi, L., Jackson, J. & Nowak, U. Giant magnetic anisotropy of the bulk antiferromagnets IrMn and IrMn₃ from first principles. *Phys. Rev. B* 79, 020403 (2009).

[2] Wang, D., Tondra, M., Nordman, C. & Daughton, J. M. Thermal stability of spin dependent tunneling junctions pinned with IrMn. *IEEE Transactions on Magnetics* 35, 2886–2888 (1999).

[3] Kang, J. et al. Current-induced manipulation of exchange bias in IrMn/NiFe bilayer structures. *Nat Commun* 12, 6420 (2021).

[4] Wang, Y.-H. et al. Impact of stray field on the switching properties of perpendicular MTJ for scaled MRAM. in 2012 International Electron Devices Meeting 29.2.1-29.2.4 (2012). doi:10.1109/IEDM.2012.6479127.

[5] Jenkins, S. et al. Magnetic stray fields in nanoscale magnetic tunnel junctions. *J. Phys. D: Appl. Phys.* 53, 044001 (2019).

[6] Bloemen, P. J. H., van Kesteren, H. W., Swagten, H. J. M. & de Jonge, W. J. M. Oscillatory interlayer exchange coupling in Co/Ru multilayers and bilayers. *Phys. Rev. B* 50, 13505–13514 (1994).

[7] Cao, Z. et al. Tuning the pinning direction of giant magnetoresistive sensor by post annealing process. *Sci. China Inf. Sci.* 64, 162402 (2021).

Q3. It is hard to see the crystallinity of the TEM in Fig. 1a and interpret the layers. It looks like the Pt is amorphous and the IrMn is sitting fairly crystalline on top of it. The bright area is the MgO, correct, it is hard to tell if it is crystalline as claimed. Please better justify the claim, “Cross-sectional TEM shows clear interfaces and crystallinity of the multilayer stack which are the essential characteristics for achieving high TMR ratio in MTJs.”

Response: We thank the reviewer for pointing out the inappropriate wording in our original description. We agree that our statement was overly strong and that the presented TEM images do not justify claiming that all layers exhibit high crystallinity. In this work, the TEM analysis mainly aims to highlight the crystallinity and interfacial characteristics of the MgO and CoFeB layers^{1,2}, as shown in Fig. R21, since these layers directly determine the TMR performance. Therefore, the imaging contrast and brightness were optimized with respect to the MgO layer.

Fig. R21 | Cross-sectional TEM image of the multilayer device structure.

Because Pt has a much higher atomic number than Mg, its electron-scattering cross-section is significantly larger, resulting in a lower image brightness under the current TEM conditions and making lattice fringes difficult to resolve. Nevertheless, based on extensive literature and material properties^{3,4,5}, Pt is known to crystallize under a variety of deposition and annealing conditions. Moreover, the IrMn layer in our sample exhibits clear lattice fringes, and given its epitaxial relation and proximity, we infer that the underlying Pt layer likely possesses a certain degree of crystallinity or partial ordering, even though its lattice is not directly resolvable in this image.

The MgO lattice fringes appear less distinct than those of the adjacent layers. This is likely due to the intentionally thin MgO barrier used in our devices to enable high-speed switching characterization with oscilloscope readout. A thinner MgO layer is more susceptible to stress relaxation or interfacial mismatch, which may reduce the visibility of lattice fringes. Nonetheless,

periodic contrast modulation is still observable, indicating that the MgO barrier remains crystalline rather than amorphous, consistent with the good TMR performance of our devices.

To provide a more accurate and balanced description of the crystallinity of our stack, we revised the text at line 105 of the manuscript:

“Cross-sectional TEM reveals continuous interfaces across the multilayer stack and clear crystalline features in the MgO barrier and adjacent CoFeB layers, which are known to play a key role in achieving high TMR in MgO-based MTJs.”

[1] Mizunuma, K. et al. MgO barrier-perpendicular magnetic tunnel junctions with CoFe/Pd multilayers and ferromagnetic insertion layers. *Appl. Phys. Lett.* 95, 232516 (2009).

[2] Ikeda, S. et al. Tunnel magnetoresistance of 604% at 300K by suppression of Ta diffusion in CoFeB/MgO/CoFeB pseudo-spin-valves annealed at high temperature. *Appl. Phys. Lett.* 93, 082508 (2008).

[3] Epitaxial Pt(111) thin film electrodes on YSZ(111) and YSZ(100) — Preparation and characterisation. *Solid State Ionics* 178, 327–337 (2007).

[4] Tolstova, Y., Omelchenko, S. T., Shing, A. M. & Atwater, H. A. Heteroepitaxial growth of Pt and Au thin films on MgO single crystals by bias-assisted sputtering. *Sci Rep* 6, 23232 (2016).

[5] Zhao, K. & Wong, H. K. Epitaxial growth of platinum thin films on various substrates by facing-target sputtering technique. *Journal of Crystal Growth* 256, 283–287 (2003).

Q4. 70% TMR is fine, but not usually considered “high”, especially for in-plane MTJs, as it is stated in the paper.

Response: We thank the reviewer for this comment and acknowledge that the wording in the manuscript was not sufficiently precise. We agree that a TMR ratio of ~70% should not be described as “high,” and instead describe the TMR value as moderate but sufficient to clearly resolve binary resistance states in the neuronal device. For synaptic devices with TMR exceeding 90%, the larger signal margin ensures a sufficient number of resistance states to support neuromorphic computing, although further optimization is still desirable to achieve even higher TMR values. The relatively “not high” TMR observed in our devices can be mainly attributed to two factors.

First, the devices were fabricated using electron-beam lithography with manually optimized etching conditions, rather than industrial-level processes. Under such conditions, edge damage is difficult to completely avoid, particularly for small device sizes^{1,2}. In line with this, larger synaptic devices in our study exhibit TMR values exceeding 90%, as shown in Fig. 1c of the manuscript, indicating that the reduced TMR primarily originates from fabrication-induced imperfections. We

expect that further process optimization, particularly at an industrial fabrication scale, would significantly improve the TMR performance. In particular, high-quality MgO barrier deposition, higher-precision lithography (such as advanced EBL or UV lithography), and low-damage etching processes would help reduce fabrication-induced defects. In addition, optimized dielectric and electrode deposition is expected to better preserve the intrinsic tunneling properties of the MTJ stack^{3,4}.

Second, in order to capture nanosecond-scale switching dynamics using an oscilloscope, the device resistance–area (RA) product must be kept sufficiently low, which necessitates a thinner effective MgO barrier. This inevitably weakens the $\Delta 1$ symmetry–filtered coherent tunneling channel, which is responsible for the highly spin-polarized transport in crystalline MgO-based MTJs, while simultaneously enhancing the contribution of interface defects, symmetry-mismatched states, and incoherent tunneling channels^{5,6}. As a result, the spin-selective tunneling efficiency is degraded and the TMR is reduced. For practical applications, a thicker MgO barrier can be employed to restore dominant $\Delta 1$ coherent tunneling and thereby achieve higher TMR values^{7,8}.

In addition, this work is primarily intended to demonstrate a new pathway for the monolithic integration of neuronal and synaptic functionalities based on EB-MTJs, thereby establishing a proof-of-concept CSNN platform. Accordingly, the emphasis of this study is placed on elucidating the feasibility and underlying physical mechanisms of such on-chip integration, rather than on comprehensive device performance optimization at this stage. Future efforts will focus on further optimization of the EB-MRAM film stack to achieve higher TMR, as well as on collaborations with semiconductor foundries to develop robust lithography and etching processes for scaled devices, with the goal of improving device uniformity and TMR in a technologically relevant regime.

In the revised manuscript, we added the following description at line 108:

“MTJs with diverse aspect ratios and lateral dimensions are fabricated, exhibiting TMR values exceeding 70%, as shown in Fig. 1b.”

[1] Zhao, W. et al. Failure Analysis in Magnetic Tunnel Junction Nanopillar with Interfacial Perpendicular Magnetic Anisotropy. *Materials* 9, (2016).

[2] Ziaur Rahaman, Sk. et al. Process-induced magnetic tunnel junction damage and its recovery for the development of spin–orbit torque magnetic random access memory. *Journal of Magnetism and Magnetic Materials* 565, 170296 (2023).

[3] Sato, H. et al. 14ns write speed 128Mb density Embedded STT-MRAM with endurance>1010 and 10yrs retention@85°C using novel low damage MTJ integration process. in 2018 IEEE

International Electron Devices Meeting (IEDM) 27.2.1-27.2.4 (2018). doi:10.1109/IEDM.2018.8614606.

[4] Yen, A., Meiling, H. & Benschop, J. EUV Lithography at Threshold of High-Volume Manufacturing. in 2018 IEEE International Electron Devices Meeting (IEDM) 11.6.1-11.6.4 (2018). doi:10.1109/IEDM.2018.8614502.

[5] Butler, W. H., Zhang, X.-G., Schulthess, T. C. & MacLaren, J. M. Spin-dependent tunneling conductance of Fe|MgO|Fe sandwiches. *Phys. Rev. B* 63, 054416 (2001).

[6] Yuasa, S. & Djayaprawira, D. D. Giant tunnel magnetoresistance in magnetic tunnel junctions with a crystalline MgO(0 0 1) barrier. *J. Phys. D: Appl. Phys.* 40, R337 (2007).

[7] Hayakawa, J., Ikeda, S., Matsukura, F., Takahashi, H. & Ohno, H. Dependence of Giant Tunnel Magnetoresistance of Sputtered CoFeB/MgO/CoFeB Magnetic Tunnel Junctions on MgO Barrier Thickness and Annealing Temperature. *Jpn. J. Appl. Phys.* 44, L587 (2005).

[8] Baez Flores, G. G., van Schilfgaarde, M. & Belashchenko, K. D. Tunneling magnetoresistance in MgO tunnel junctions with Fe-based leads in empirically corrected density functional theory. *Phys. Rev. B* 110, 224425 (2024).

Q5. Please clarify if the long axis of the ellipse is along y, the field annealing direction, and that the long axis is opposite the SOT current direction axis (x). It would be helpful to show in Fig 1a where the voltage V_{eff} is applied across the Pt heavy metal underneath, and have the figure actually show how the anisotropy is on the track, right now the angled view of the cartoon doesn't show that.

Response: We appreciate the reviewer's comment regarding the clarity of the device geometry and schematic representation. In our devices, the long axis of the elliptical MTJ is aligned along the x direction, which is also the direction of the SOT current flowing in the underlying Pt heavy-metal layer, as shown in Fig. R22. The short axis lies along the y direction, corresponding to the field-annealing direction; therefore, the magnetization easy axis of the device is oriented along the short axis of the ellipse.

This geometry corresponds to a y-type SOT writing configuration, in which aligning the short axis of the ellipse along the y direction allows the bottom electrode width to be reduced. As a result, a higher current density can be achieved under the same applied current and it can significantly lower the switching energy consumption. For clarity, we label the magnetization directions of the free layer and the antiferromagnetic layer, as well as the magnitude and polarity of the effective SOT voltage applied to the Pt electrode. When the SOT current flows along the $\pm x$ direction, spin-polarized electrons with opposite spin orientations accumulate at the top and bottom surfaces of the Pt layer, respectively, due to the spin Hall effect^{1,2}. The resulting spin-orbit

torque induces a switching of both the ferromagnetic and antiferromagnetic moments, leading to a deterministic final magnetization configuration defined by the SOT polarity.

Fig. R22 | Schematic model of the device layer structure.

In the revised manuscript, Fig. 1a has been updated to explicitly indicate the direction of the applied SOT current across the Pt heavy-metal layer and the orientations of both the exchange bias and the magnetization of the free layer. To complement this update, the corresponding textual description has also been added at line 102:

“This procedure aligns the magnetization of the device along the -y direction, consistent with the spin polarization generated by the SOT current flowing along the x direction in the underlying Pt electrode.”

Additionally, the viewing angle of the schematic has also been adjusted to ensure that the anisotropy directions are clearly visible and unambiguous.

[1] Sinova, J., Valenzuela, S. O., Wunderlich, J., Back, C. H. & Jungwirth, T. Spin Hall effects. *Rev. Mod. Phys.* 87, 1213–1260 (2015).

[2] Hoffmann, A. Spin Hall Effects in Metals. *IEEE Transactions on Magnetics* 49, 5172–5193 (2013).

Q6. Fig. 2a is overall very interesting that the many stable states can be achieved across the TMR range. But, MTJ based devices currently have the challenge of small on/off ratio relative to other technologies. This gives narrower room for each level across devices. Can you show Fig. 2a for more than one nominally identical device and see the spread in each R state? That will give a better idea how many states you actually can resolve, it is probably smaller than 25. E.g. see Fig. 7 of DOI 10.1109/IEDM45625.2022.10019564 .

Response: We thank the reviewer for raising the important question regarding the separability of resistance states, which is highly relevant for defining distinct synaptic weights and for improving

the overall network accuracy. We agree that neuromorphic computing based on MTJ devices generally suffers from a relatively small on/off ratio and limited resistance-state distinguishability. In this context, Fig. 2a in our manuscript is intended to demonstrate the maximum number of resistance states achievable in a single device under single-shot programming, serving as a characterization of intrinsic device capability rather than the effective number of reliably distinguishable states.

To quantitatively evaluate the separability of different resistance states, we performed repeated reset–set measurements on multiple devices using 100 cycles of 0.4-ns SOT pulses. For each programming voltage, we extracted the mean value and standard deviation of the normalized resistance states. Fig. R23 shows a representative example of the normalized resistance states and their corresponding standard deviations as a function of the applied voltage.

Fig. R23 | Schematic illustration of the resistance states and error bars as a function of SOT pulse voltage.

Following the representation used in Fig. 7 of Agrawal et al.¹, we selected a subset of resistance states along the switching curve that exhibit relatively good separation and modeled their resistance distributions using Gaussian functions:

$$f(x) = \exp\left[-\frac{(x - \mu)^2}{2\sigma^2}\right]$$

where μ denotes the mean value of the normalized resistance state and σ represents the standard deviation. The resulting resistance distributions are shown in Fig. R24.

Fig. R24 | Gaussian distributions of different intermediate states of the normalized device resistance.

As correctly pointed out by the reviewer, due to the limited switching ratio and the presence of write variability, the number of effectively distinguishable resistance states is smaller than the maximum value of 25 shown in Fig. 2a, and is approximately 15 in practice. To provide a more comprehensive picture, we repeated the same analysis on ten different devices. The statistical distribution of distinguishable resistance states is summarized in Fig. R25, showing that the number of separable states lies in the range of 11–15 across these devices.

Fig. R25 | Gaussian distributions of different intermediate states of normalized resistance for multiple devices.

To assess the effect of a reduced number of synaptic resistance states on network performance, we first decreased the number of states in the full EB-MTJ CSNN from 25 (as in the main text) to 15 and then simulated the corresponding network behavior, as shown in Fig. R26. The results show that the network with 15 states exhibits only a marginal reduction in maximum accuracy, from 96.2% to 94.8%, throughout the training process. Overall, this demonstrates that the network is largely robust to variations in the number of synaptic states, indicating that the current synaptic devices are sufficient to support high-precision gesture recognition.

Fig. R26 | Effect of different numbers of synaptic resistance states on network accuracy

Although this slight reduction in state number has minimal impact on performance, future work will focus on improving the separation between resistance states to increase the number of effectively distinguishable states. Notably, the observed reduction in distinguishable states is primarily attributed to two factors.

First, the TMR ratio of the devices is not sufficiently large, which is related to both the film structure and fabrication-induced damage. In order to directly observe nanosecond-scale switching dynamics using a high-speed oscilloscope for implementing the LIF neuron function, the device resistance could not be excessively high. Consequently, a relatively thin MgO barrier (1.5 nm) was employed, which limits coherent tunneling and reduces the achievable TMR^{2,3}. In future device designs aimed at practical applications, thicker MgO barriers can be adopted to significantly enhance the TMR and improve resistance-state separation. In addition, the devices were fabricated using laboratory-scale EBL lithography and etching with manually optimized parameters, which may introduce edge damage and further degrade the TMR^{4,5}.

Second, device-to-device and state-to-state variability also limits resistance separability. The distinguishability of resistance states is jointly determined by the spacing between intermediate resistance levels and their associated fluctuations. As discussed above, the laboratory-scale fabrication process inevitably leads to variations in device dimensions, morphology, and interface roughness, resulting in relatively large write-induced fluctuations and non-uniform error bars across different resistance states. As a result, some intermediate states become obscured by the overlap of neighboring distributions. In future neuromorphic device arrays, fabrication in a high-cleanliness semiconductor foundry with mature and well-controlled processes is expected to substantially reduce both intra-device and inter-device variability, thereby increasing the number of reliably distinguishable resistance states.

To complement the discussion on how the separation between multiple resistance states and the number of resistance levels influence the computational accuracy of the CSNN, the revised manuscript now includes a brief description at line 157:

“Further measurements confirm that multi-level operation is maintained up to pulse widths of 5 ns (typically >20 levels), with the resistance levels remaining clearly distinguishable, enabling reliable multi-state operation (details are provided in Section 5 and 6 of the SI).”

The comprehensive analysis is presented in *Section 5 and 6 of the SI*.

- [1] Agrawal, V. et al. Subthreshold operation of SONOS analog memory to enable accurate low-power neural network inference. in 2022 International Electron Devices Meeting (IEDM) 21.7.1-21.7.4 (2022). doi:10.1109/IEDM45625.2022.10019564.
- [2] Butler, W. H., Zhang, X.-G., Schulthess, T. C. & MacLaren, J. M. Spin-dependent tunneling conductance of Fe|MgO|Fe sandwiches. *Phys. Rev. B* 63, 054416 (2001).
- [3] Yuasa, S. & Djayaprawira, D. D. Giant tunnel magnetoresistance in magnetic tunnel junctions with a crystalline MgO(0 0 1) barrier. *J. Phys. D: Appl. Phys.* 40, R337 (2007).
- [4] Zhao, W. et al. Failure Analysis in Magnetic Tunnel Junction Nanopillar with Interfacial Perpendicular Magnetic Anisotropy. *Materials* 9, (2016).
- [5] Ziaur Rahaman, Sk. et al. Process-induced magnetic tunnel junction damage and its recovery for the development of spin-orbit torque magnetic random access memory. *Journal of Magnetism and Magnetic Materials* 565, 170296 (2023).

Q7. For Fig. 2f, usually a linear/symmetric pulse train is expected shown to see the linearity and symmetry over more than one ramp up/down.

Response: We thank the reviewer for the suggestion to further examine the cycle-to-cycle stability of the resistance evolution. In line with the reviewer’s suggestion, we performed additional measurements on the elliptical synaptic device over ten consecutive up/down programming cycles, as shown in Fig. R27. The upper panel displays the effective SOT pulse voltage V_{eff} applied to the device with a pulse width of 0.4 ns, while the lower panel shows the corresponding TMR values read out by dc measurements after each writing pulse. Specifically, the blue curves correspond to the TMR evolution under negative SOT voltages with gradually increasing amplitude, whereas the orange curves correspond to positive SOT voltages. As expected, within the writing threshold window, the device resistance evolves progressively with increasing pulse amplitude, exhibiting a quasi-linear dependence, consistent with the behavior reported in Fig. 2f of the manuscript.

Fig. R27 | Schematic of multiple up/down cycles of the device linear response to SOT pulse voltage.

Notably, across the ten consecutive increasing/decreasing cycles, the device shows good symmetry and stability during the write process. The resistance values of both the AP and P states remain nearly unchanged, indicating good repeatability and robustness of the multi-level switching behavior.

In the revised manuscript, Fig. 2f has been revised to show ten consecutive up/down switching cycles instead of a single cycle, demonstrating the reproducibility and stability of the synaptic linearity. The updated data are shown in the revised Fig. 2f of the main text and are also provided here as Fig. R27. Correspondingly, the description in the revised manuscript has been updated at line 234:

“As shown in Fig. 2f, by applying a series of SOT pulses with gradually increasing amplitude, the synaptic device exhibits a linear resistance response to the pulse voltage. Notably, this behavior is reproduced over ten consecutive programming cycles, demonstrating the stability and high reproducibility of the analog switching characteristics, which is essential for applications in neuromorphic computing.”

Q8. The Fig 2c data was not clear, how is field immunity up to 2 T shown by the data? How does this compare to Fig. 1b where it is assumed a bigger field is used to set the device in the n- or n+ states?

Response: We thank the reviewer for raising this important point and apologize for the unclear presentation in the original manuscript. Regarding Fig. 2c, the magnetic-field immunity up to 2 T refers to the retention of the resistance state after the removal of an external magnetic field, rather than the absence of resistance change under an applied field¹. In the measurement shown in Fig. 2c, a perpendicular magnetic field (up to 2 T) is applied to the device and subsequently removed. Although the device resistance varies during field application due to the gradual canting of the free and reference layer magnetizations toward the field direction, the resistance always returns to

the same initial value once the field is removed, independent of the maximum field amplitude. This recovery is enabled by the antiferromagnetic IrMn layer, whose exchange bias is insensitive to the external magnetic field and remains pinned along the $\pm y$ direction. Upon field removal, the free and reference layers relax back to their original orientations under the combined effect of the IrMn exchange bias and the SAF RKKY coupling, thereby restoring the original resistance state²⁻⁴. This field-removal invariance demonstrates the magnetic-field robustness up to 2 T.

Regarding the apparent comparison with Fig. 1b, we clarify that the magnetic field in Fig. 1b is not used to set the n^+ or n^- states. Instead, these states are defined by the exchange bias direction of the bottom IrMn layer, which can only be switched by SOT-induced writing and is insensitive to external magnetic fields. The R - H curves in Fig. 1b merely illustrate the field response of the device after the state has been set by SOT current. Importantly, once the external field is removed, the zero-field resistance always relaxes back to the state defined by the IrMn exchange bias, even when the applied field previously exceeded the saturation field.

To further clarify this distinction, we have added R - H curves measured under x- and y-direction magnetic fields up to 500 mT (well above the saturation field), together with the corresponding field-robustness data, as shown in Fig. R28.

Fig. R28 | Magnetoresistance characterization of the device. **a** R - H curve under large magnetic field applied along the y direction. **b** Resistance variation when $\pm y$ magnetic fields are applied and removed, showing the device's resilience to y-direction fields. **c** R - H curve under large magnetic field applied along the x direction. **d** Resistance response to $\pm x$ magnetic fields, demonstrating the device's resilience to x-direction fields.

To clearly illustrate the magnetic-field robustness of the devices and the underlying physical mechanism, we have revised line 163 of the main text as follows:

“Furthermore, Fig. 2c shows that although the device resistance varies under the application of a perpendicular magnetic field, it fully recovers to its original value after field removal, even at fields up to 2 T (details are provided in Section 8 of the SI). This field-removal invariance, together with the robust thermal stability, confirms that the EB-MTJ devices maintain stable and discrete memristive states against thermal and magnetic perturbations.”

Additionally, R–H measurements under x- and y-direction magnetic fields up to 500 mT have been added to *Section 8 of the SI*, showing that the zero-field resistance consistently returns to its original value, further confirming the magnetic-field robustness of the devices.

[1] Du, A. et al. Electrical manipulation and detection of antiferromagnetism in magnetic tunnel junctions. *Nature Electronics* 6, 425–433 (2023).

[2] Peng, S. et al. Exchange bias switching in an antiferromagnet/ferromagnet bilayer driven by spin–orbit torque. *Nat Electron* 3, 757–764 (2020).

[3] Power, S. R., Ferreira, M. S., Power, S. R. & Ferreira, M. S. Indirect Exchange and Ruderman–Kittel–Kasuya–Yosida (RKKY) Interactions in Magnetically-Doped Graphene. *Crystals* 3, 49–78 (2013).

[4] Ma, M. et al. Optical control of RKKY coupling and perpendicular magnetic anisotropy in a synthetic antiferromagnet. *Nat Commun* 16, 4401 (2025).

Q9. The pulses applied are very narrow (e.g. 0.4 ns) for the elliptical devices, can the authors comment on the shape of the pulse and if any reflections are expected? Was a ground/signal/ground setup needed? What is the behavior if the pulse is longer (~1-5 ns)?

Response: We thank the reviewer for the thoughtful question concerning the pulse characteristics and measurement configuration. We investigated possible pulse attenuation and reflection effects during pulse transmission using the setup shown in Fig. R29a. A Keysight M8190A arbitrary waveform generator (maximum sampling rate of 8 GHz) was used to generate the pulses, which were amplified and then applied to the bottom electrode of the device. The device shown in the center of the setup is the GSG-compatible structure used in this work. One side of the device was connected to the waveform generator via a GSG probe, while the other side was connected to a high-resolution oscilloscope through another GSG probe for signal readout. Below we address the pulse shape and possible reflections and the pulse-width dependence of the switching behavior in turn.

Fig. R29 | Characterization of the input pulse waveform. **a** Schematic of the measurement setup for pulse waveform characterization. **b** Comparison of the applied pulse and the pulse measured after passing through the metal bottom electrode (MBE), i.e. the SOT channel of the device.

As shown by the black curve in Fig. R29b, a 0.4-ns SOT pulse with a nominal amplitude of 1 V after amplification was directly measured from the waveform generator output. The rise and fall times of the pulse are both approximately 160 ps. The red curve in Fig. R29b shows the pulse waveform measured after transmission through the device bottom electrode. Compared with the directly measured waveform, the rising edge remains nearly unchanged, whereas the falling edge is slightly extended. Nevertheless, the overall pulse width shows no significant variation. This can be understood as a result of the effective RC effect introduced by the bottom electrode^{1,2}. While the leading edge is dominated by the initial fast excitation and remains largely unaffected, the high-frequency components contributing to the trailing edge are partially attenuated, resulting in a prolonged decay. As a consequence, the pulse shape becomes slightly asymmetric, while the full pulse width remains nearly unchanged and does not affect the switching behavior of the device.

Since both the output impedance of the arbitrary waveform generator and the input impedance of the oscilloscope are 50 Ω , while the resistance of the device bottom electrode is 382.8 Ω , the expected voltage division yields $V_{\text{direct}} = 50/(50+50)*V_{\text{input}} = 0.5V_{\text{input}}$ and $V_{\text{MBE}} = 50/(50+50+382.8)*V_{\text{input}} = 0.1035V_{\text{input}}$, such that $V_{\text{MBE}} = 0.207V_{\text{direct}}$. Experimentally, the measured voltages are 1.007 V and 0.171 V, respectively, corresponding to $V_{\text{MBE,real}} = 0.170V_{\text{direct,real}}$. The deviation between the expected and measured values is approximately 17.9%, which we attribute mainly to impedance mismatch between the device bottom electrode and the 50 Ω measurement system. Nevertheless, under ultrashort pulse conditions (0.4 ns), this level of attenuation is within an acceptable range.

To provide a detailed account of the voltage attenuation and distortion applied to the devices, we have revised the main text at line 117 of the revised manuscript:

“Moreover, the resistance state of the EB-MTJ can be modified by applying write voltage pulses through the bottom electrode, with pulse widths down to 0.4 ns as shown in Figs. 1c and 1d. The

detailed pulse shapes and measurement procedures are described in the Methods and in Section 1 and 2 of the Supplementary Information (SI).”

The corresponding discussion and additional details have also been included in Section 1 of the Supplementary Information.

Moreover, to provide further insight into the device switching behavior at different pulse widths, we have supplemented the SOT switching characteristics of the elliptical synaptic devices for pulse widths ranging from 0.4 to 5 ns, as shown in Fig. R30. Clear multilevel resistance behavior is observed across this entire pulse-width range. Taking a TMR difference greater than 2% as the criterion for distinguishing resistance states, we counted the number of accessible resistance levels at different pulse widths and summarized the results in Fig. R30f.

Fig. R30 | Multilevel resistance characteristics of the device under different pulse widths. **a–e** R–V switching curves of the device under SOT pulses with widths of 0.4–5 ns. **f** Statistics of the number of resistance states at different pulse widths.

Although the number of resistance states decreases with increasing pulse width, more than 20 distinct resistance states are still achieved for all pulse widths between 0.4 and 5 ns, indicating robust multilevel characteristics even over a wide pulse-width range. Notably, for the 0.4-ns pulses, the full multilevel writing window spans from 1.38 to 1.94 V, corresponding to approximately 33.5% of the critical switching threshold defined at $(R_{AP}+R_P)/2$. This relatively wide programming margin enables reliable and fine control of the device resistance during ultrafast programming.

To clarify the pulse-width dependence of the synaptic device switching behavior and the corresponding switching-voltage range, we have added the following description in the revised manuscript at line 157:

“Further measurements confirm that multi-level operation is maintained up to pulse widths of 5 ns (typically >20 levels), with the resistance levels remaining clearly distinguishable, enabling reliable multi-state operation (details are provided in Section 5 and 6 of the SI).”

The corresponding data have also been included in *Section 5 of the Supplementary Information*.

[1] Schuster, C. & Fichtner, W. Parasitic modes on printed circuit boards and their effects on EMC and signal integrity. *IEEE Transactions on Electromagnetic Compatibility* 43, 416–425 (2001).

[2] Chang, C. T. M., Namordi, M. R. & White, W. A. The effect of parasitic capacitances on the circuit speed of GaAs MESFET ring oscillators. *IEEE Transactions on Electron Devices* 29, 1805–1809 (1982).

Q10. For the neural network application, can the authors add more details about how the device data is input into the model? Expand on the line, “All neurons were modeled using parameters derived from the thermally assisted switching dynamics of EB-MTJs.”

Response: We appreciate the reviewer’s suggestions for additional detail. In our simulations, device measurements are not used only to “fit an STDP curve”; rather, multiple experimentally extracted EB-MTJ characteristics are mapped into the network model at both the synapse and neuron levels, so that network dynamics reflect thermally assisted switching physics.

Device-to-network mapping for synaptic weights (25 stable states)

First, the measured multi-level resistance states of EB-MTJs are used to store network weights. Specifically, the EB-MTJ provides 25 stable resistance states, and all synaptic weights in both convolutional layers and fully connected layers are quantized to these 25 discrete levels via a deterministic mapping $w \leftrightarrow \text{TMR}$.

The TMR sequence we used for storing synaptic weight is [0, 0.0116, 0.03047, 0.08488, 0.10981, 0.12656, 0.15316, 0.19341, 0.21864, 0.23759, 0.3133, 0.37014, 0.41417, 0.41773, 0.50603, 0.53217, 0.5997, 0.63554, 0.62743, 0.62941, 0.64062, 0.66806, 0.77575, 0.85836, 0.90784]. To validate the advantage of the device’s multi-level capability, we compared network training using 5-level, 15-level, and 25-level quantization; the accuracy curves show that 25-level quantization consistently outperforms lower-level quantization, demonstrating the benefit of the device’s 25 stable states for weight storage.

Fig. R31 | Comparison between real weights and quantized weights in our neural network. **a,b** The real weights and quantized weights in our neural network. The input index and output index denote neuron indices in the previous and next layer, respectively; each pixel is the weight $W_{j,i}$ connecting input neuron i to output neuron j . **c** The difference ($W_{\text{quant}} - W_{\text{real}}$) of the real weights and quantized weights.

To stabilize training under quantization, we adopt quantization-aware training (QAT)¹, a gradual quantization schedule:

$$w_{\text{eff}} = (1 - \alpha) w_{\text{real}} + \alpha \text{Quant}(w_{\text{real}}),$$

where w_{real} is the real-valued weight updated by learning, $\text{Quant}(\cdot)$ maps w_{real} to the nearest of the 25 TMR levels of EB-MTJs, and α is smoothly increased from 0 to 1 during training. Thus, $\alpha = 0$ corresponds to fully real-valued training, while $\alpha = 1$ corresponds to fully device-quantized weights. This interpolation allows the network to converge stably before being fully constrained by device discreteness.

Fig. R32 | a The test accuracy of different numbers of states. **b** QAT controlled parameter α changed from 0 to 1 within 20 epochs.

Device-to-neuron mapping from thermally assisted switching dynamics (integration, leak, firing)

Second, each EB-MTJ neuron is modeled using parameters derived from measured thermally assisted dynamics, rather than assuming an abstract LIF neuron. Under input current pulses from

pre-neurons, the local temperature T evolves according to the device thermal accumulation and relaxation laws²:

$$T_{post} = \left(\frac{J_i^2 \alpha}{K} + T_0 \right) (1 - \alpha) + T_{pre} \alpha, \quad \alpha = e^{-\tau/\tau_0},$$

and during current cessation,

$$T_{post} = T_0(1 - \alpha) + T_{pre} \alpha,$$

where J_i is the input current density, K is the combined specific heat/heat generation factor, T_0 is ambient temperature, and τ_0 is the device thermal response time. These equations explicitly implement integration (temperature accumulation) and leak (exponential thermal relaxation), making $T(t)$ the physical analogue of membrane potential. A firing event is triggered when successive spikes raise $T(t)$ above a threshold T_{th} , at which point the EB-MTJ switches state (representing threshold crossing). In the neural model, the effective leak constant and firing threshold are therefore derived from experimentally measured thermal relaxation traces and switching-current transfer curves (e.g., extracting τ_0 and the temperature-dependent switching threshold statistics).

For the STDP training part, we employ the standard exponential pair rule:

$$\Delta w(\Delta t) = \begin{cases} A_+ \exp(-\Delta t/\tau_+) & \Delta t > 0, \\ -A_- \exp(+\Delta t/\tau_-) & \Delta t < 0, \end{cases}$$

where $\Delta t = t_{post} - t_{pre}$. The parameters A_{\pm} and τ_{\pm} are obtained by fitting experimentally measured weight-change (or resistance-change) curves under paired pre/post stimulation protocols. In our device fitting, we obtained $\tau_+ = 3.81\text{ns}$, $\tau_- = 2.70\text{ns}$, $A_2 = 0.787$, $A_2^+ = 0.789$ and incorporated them into our STDP training process.

Fig. R33 | STDP rule of long-term plasticity demonstrated for paired spikes

To simplify our training process and avoid the needs to record all firing times of pre and post neurons, we use the trace method to implement STDP in our neural network as depicted in Section 16 of the SI. The trace-based formulation of STDP:

$$tr_{pre}[i](t) = tr_{pre}[i](t-1)e^{-\frac{1}{\tau_{pre}}} + s_i(t)$$

$$tr_{post}[i](t) = tr_{post}[i](t-1)e^{-\frac{1}{\tau_{pre}}} + s_j(t)$$

$$\Delta W_{ij}(t) = F_{pot}(W_{ij}(t))tr_{pre}[i](t)s_j(t) - F_{dep}(W_{ij}(t))tr_{post}[j](t)s_i(t)$$

where $s_{i,j}(t) \in \{0,1\}$ are spike indicators and $F_{pot/dep}$ are the bounded update functions. The chosen decay constants $\tau_{pre} = 2$ and $\tau_{post} = 4$ match the intrinsic relaxation of our EB-MTJ synapses τ_+ and τ_- .

To provide a detailed explanation of how our device data are input into the neural network model and how the neuronal device undergoes thermally assisted state switching, we have added the following description in the revised manuscript (line 313):

“..., with the mapping of device-level parameters to the neuron model described in the Methods, and the detailed simulation procedures described in Section 11 of the SI.”

And provided a more detailed explanation in the Methods section (line 467):

“All neurons were modeled using parameters derived from the thermally assisted switching dynamics of EB-MTJs. Specifically, when presynaptic current pulses are applied, the local device temperature T follows thermal accumulation with exponential memory, which we implement in discrete form as

$$T_{post} = \left(\frac{J_i^2}{K} + T_0\right)(1 - \alpha) + \alpha T_{pre}, \alpha = e^{-\tau/\tau_0},$$

and during pulse cessation (no Joule heating),

$$T_{post} = T_0(1 - \alpha) + \alpha T_{pre}.$$

Here J_i is the input current density, K is an effective thermal factor capturing heat capacity and heat generation, T_0 is the ambient temperature, τ_0 is the device thermal response time, and τ is the pulse interval. These equations explicitly realize integration (temperature accumulation under repeated pulses) and leak (exponential relaxation toward T_0), making $T(t)$ the physical state variable analogous to the membrane potential in an LIF neuron.”

The corresponding analyses are provided in *Section 11 of the Supporting Information*.

[1] Krestinskaya, L. Zhang and K. N. Salama, "Towards Efficient RRAM-based Quantized Neural Networks Hardware: State-of-the-art and Open Issues," *2022 IEEE 22nd International Conference on Nanotechnology (NANO)*, Palma de Mallorca, Spain, 2022, pp. 465-468, doi: 10.1109/NANO54668.2022.9928590.

[2] D. Zhu *et al.*, "Thermally Driven Leaky-Integrate-and-Fire Spintronic Neurons With Stray-Field-Enabled Self-Reset for Neuromorphic Computing," in *IEEE Electron Device Letters*, vol. 46, no. 8, pp. 1437-1440, Aug. 2025, doi: 10.1109/LED.2025.3580276.

Q11. Please check grammar throughout, here are some examples:

- o E.g. “on-chip co-integrated memristive synapse and leaky-integrate-fire (LIF) neuron designed..” should have “an” before
- o Verb missing on line 47
- o “we figures out” line 144

Response: We thank the reviewer for pointing out these grammatical and typographical errors. All the issues mentioned (including the missing article, missing verb, and incorrect verb tense) have been carefully corrected. In addition, the entire manuscript has been thoroughly proofread and revised to improve grammatical accuracy and clarity throughout. We have also refined several definitions and standardized terminology to improve clarity and consistency.

Q12. If the nonuniform grain size of the IrMn is the source of multiple levels, then can the authors make any conclusions about the best way to grow the material for this application?

Response: We thank the reviewer for the constructive suggestion regarding the optimization of thin-film growth conditions and process design. This comment provides valuable guidance for our future efforts toward further improving film quality and device performance. Our multilevel states are not instantaneous resistance values measured under different bias or temperature conditions, but retained magnetic configurations written by current pulses and read out under the same small sensing current. Their physical origin lies in which microscopic magnetic entities can be reconfigured during a write pulse and remain stable after stimulus removal. In polycrystalline exchange-bias systems, the antiferromagnet consists of an ensemble of grains, each acting as a thermally activated unit characterized by an energy barrier $\Delta E \approx K_{AF} \cdot v$ and a blocking temperature T_B determined by its grain volume v . A distribution of grain volumes therefore produces a distribution of blocking temperatures and activation thresholds, governing thermal stability, training effects and partial exchange-bias reconfiguration in IrMn-based systems¹⁻⁵. In our device, SOT together with Joule heating transiently lowers local pinning barriers and enables partial ferromagnetic reversal, while only those IrMn grains whose blocking temperatures fall within the accessible thermal window can reorient and subsequently lock the reversed configuration. Consequently, the IrMn grain-size (or grain-volume) distribution is the key materials parameter that controls how many distinct and reproducible intermediate states can be stabilized.

The “best way to grow IrMn” is therefore to engineer a controlled ensemble of relatively small grains with a useful spread of blocking temperatures. Grains that are too small become thermally unstable, whereas excessively large grains remain inert under practical write stimuli and reduce the number of accessible levels. The desired microstructure is thus one whose $T_{B(v)}$ distribution overlaps the temperature window reached during write pulses while maintaining

sufficiently strong and stable exchange bias. Such trade-offs among grain size, texture, annealing and exchange-bias stability have been widely reported in IrMn/CoFeB and related systems^{3,7}.

From a thin-film growth perspective, increasing Ar pressure or flow enhances gas-phase scattering and favors finer grains, as described by classical structure-zone models^{6,8}, and pressure-controlled microstructural tuning has been widely used to modify magnetic properties in IrMn-containing stacks¹. Increasing the target–substrate distance further suppresses grain coarsening by reducing the effective energy flux delivered to the growing film. Sputter power and deposition rate must be treated cautiously because they modify both energy flux and surface mobility; IrMn studies show that growth kinetics can systematically change median grain size and, in some regimes, higher growth rates correlate with larger IrMn grains as directly observed by TEM⁷.

Annealing process establishes exchange bias field and improves interfacial properties but also promotes grain growth and interdiffusion at the AFM/FM interface. In IrMn/CoFeB systems, annealing modifies texture, roughness and Mn diffusion/dilution, leading to changes in both exchange-bias magnitude and stability^{3,7}, while higher annealing temperatures in Mn–Ir films increase lateral grain size⁷. For multilevel EB-MTJs, overly aggressive anneals are therefore undesirable; instead, a moderate thermal budget is preferred to establish robust exchange bias while preserving a useful spread of blocking temperatures.

Taken together, IrMn growth for multilevel EB-MTJs should be steered toward relatively small grains with a controlled but finite distribution width by combining higher effective scattering conditions, appropriate power and rate settings that avoid high-mobility coarsening, and a moderate annealing budget that establishes exchange bias without collapsing the grain ensemble. With the present nanometric IrMn microstructure, we obtain about 25 stable levels in ~100-nm devices.

In the revised manuscript, we have added a forward-looking discussion at line 346 on potential strategies to achieve a larger number of more stable multilevel states in in-plane FM/AFM systems, aiming to further enhance device performance. The added text reads:

“..., and suggest that future improvements in capacity could be realized through antiferromagnetic microstructure engineering—by reducing the effective grain size and tailoring the blocking-temperature distribution, even finer and more numerous intermediate states may be stabilized.”

[1] Vallejo-Fernandez, G., Fernandez-Outon, L. E. & O’Grady, K. Measurement of the anisotropy constant of antiferromagnets in metallic polycrystalline exchange biased systems. Appl. Phys. Lett. 91, 212503 (2007).

- [2] Nogués, J. & Schuller, I. K. Exchange bias. *Journal of Magnetism and Magnetic Materials* 192, 203–232 (1999).
- [3] Keller, J. et al. Domain state model for exchange bias. II. Experiments. *Phys. Rev. B* 66, 014431 (2002).
- [4] Du, A. et al. Electrical manipulation and detection of antiferromagnetism in magnetic tunnel junctions. *Nature Electronics* 6, 425–433 (2023).
- [5] Bai, Y. et al. Ru/IrMn Interfacial Orbital-to-Spin Conversion for Antiferromagnetic Switching in Magnetic Tunnel Junctions. *Nano Lett.* 25, 14843–14849 (2025).
- [6] Thornton, J. A. Influence of apparatus geometry and deposition conditions on the structure and topography of thick sputtered coatings. *J. Vac. Sci. Technol.* 11, 666–670 (1974).
- [7] Raju, M., Chaudhary, S. & Pandya, D. K. Magnetic annealing of the ion-beam sputtered IrMn/CoFeB bilayers – positive exchange bias and coercivity behaviour. *Eur. Phys. J. B* 86, 491 (2013).
- [8] Movchan, B. A. & Demchishin, A. V. STRUCTURE AND PROPERTIES OF THICK CONDENSATES OF NICKEL, TITANIUM, TUNGSTEN, ALUMINUM OXIDES, AND ZIRCONIUM DIOXIDE IN VACUUM. *Fiz. Metal. Metalloved.* 28: 653-60 (Oct 1969). (1968).

Response To the Reviewer #3

Q1. The authors demonstrated various TMR values for the exchange bias MTJ device to show its multistate synaptic behavior. Since TMR values are highly sensitive to both temperature and voltage, a physics-based justification supported by equations is necessary to explain how the EB-based mechanism achieves the discrete 25 states shown in Figure-2. The provided Equations-1 and 2 are too generic in this context. In addition, the combined influence of temperature and voltage on TMR need be analyzed and discussed.

Response: We thank the reviewer for this question and appreciate the opportunity to clarify the origin of the intermediate resistance states. We note that the approximately 25 intermediate states shown in Fig. 2a of the main text are measured under identical and low readout conditions. Specifically, all resistance states are read using the same small sensing current of 1 μA , corresponding to a read voltage of approximately 31.3 mV. The measurement protocol in Fig. 2a of the main text follows a well-defined write–read sequence, which is illustrated in Fig. R34. Specifically, a single write pulse with a duration of 0.4 ns is first applied to the device. After the write operation is completed, a waiting time of approximately 300 ms is introduced before the resistance is read. The resistance is then measured using the small sensing current of 1 μA for a duration of about 500 ms. After an additional interval of approximately 1 s, the next write pulse is

applied, and the same write-read sequence is repeated. As a result, the intermediate resistance levels observed in Fig. 2a correspond to retained and well-defined device states established during the write operation, rather than to transient changes during readout.

Fig. R34 | Timing diagram of SOT writing pulses and DC read current during SOT-induced switching of device.

Given this measurement sequence, the write process has fully terminated well before the readout begins, and the temperature rise induced by the $0.4 ns$ write pulse is expected to relax back to a level close to room temperature within the $300 ms$ interval, such that no appreciable thermal accumulation remains during the read operation. Importantly, all resistance states are read using the same small sensing current of $1 \mu A$. At this read current, both the temperature increase due to Joule heating and the effect of the read voltage on the TMR are minimal. Fig. R35 shows the dependence of the device resistance on the applied DC read current. From the fitting, we estimate that, compared with the resistance measured without current injection, the AP-state resistance decreases by only $\sim 1.02\%$ when a read current of $1 \mu A$ (corresponding to a read voltage of $31.3 mV$) is applied, corresponding to a TMR variation of less than 0.9% . Therefore, the DC read current of $1 \mu A$ employed in our measurements has a negligible impact on the resistance states of the device.

Fig. R35 | Variation of the AP-state resistance of the device under different DC read currents.

Under these conditions, the resistance measured during readout reflects a retained and well-defined device state, and the intermediate resistance states observed in Fig. 2a in the main

text can therefore be reasonably attributed to intrinsic magnetic configurations determined by the spatial distribution and proportion of reversed magnetic domains, rather than to transient variations in resistance caused by read-induced heating or read voltage.

Next, we describe the physical origin of the multilevel resistance states in our devices and the underlying physical picture.

In conventional spin-orbit torque magnetic tunnel junctions (SOT-MTJs) with a purely ferromagnetic free layer, the resistance switching behavior is intrinsically binary. This is because the switching of the ferromagnetic magnetization is a collective process: once the SOT-induced effective field exceeds the critical threshold, magnetic domains rapidly expand and coalesce, driving the system toward a complete switching. Although multi-domain configurations may transiently appear during switching, these states are energetically unstable. Under continued current excitation or thermal fluctuations, the system inevitably relaxes into one of the two energetically favored states, resulting in binary resistance characteristics.

The emergence of stable intermediate resistance states in our devices arises from the introduction of an antiferromagnetic layer within the free layer, which fundamentally alters the switching mechanism. Antiferromagnets such as $\text{Ir}_{20}\text{Mn}_{80}$ naturally exhibit a distribution of grain sizes, as shown in Fig. R36 (corresponding to Fig. 2d in the main text). The grain-size distribution can be well described by a log-normal function, which can be expressed as¹

$$f(v)dv = \frac{1}{\sqrt{2\pi}\sigma v} \exp\left[-\frac{(\ln(v) - \mu)^2}{2\sigma^2}\right] dv$$

where v is the grain size and $\ln(v)$ is normally distributed with a mean value μ and a standard deviation σ . The fit parameters are $\mu = 2.04$ and $\sigma = 0.190$ for the $\text{Ir}_{20}\text{Mn}_{80}$ single layer (yellow), and $\mu = 2.14$ and $\sigma = 0.199$ for the sandwiched structure (green), corresponding to median grain sizes of 7.71 nm and 8.47 nm, respectively.

Fig. R36 | Grain size distributions in the $\text{Ir}_{20}\text{Mn}_{80}$ and the $\text{Co}_{20}\text{Fe}_{60}\text{B}_{20}/\text{Ir}_{20}\text{Mn}_{80}/\text{Co}_{20}\text{Fe}_{60}\text{B}_{20}$ multilayer structure.

Larger antiferromagnetic grains possess sufficient anisotropy energy to remain magnetically stable at room temperature, with their blocking temperature determined by the thermal activation condition:

$$T_B(v) = K_{AF} * v / [k_B * \ln(\tau_m / \tau_0)].$$

where K_{AF} is anisotropy energy density of the antiferromagnet, k_B is the Boltzmann constant, τ_m is measurement time, and τ_0 is attempt time. Through interfacial exchange coupling, these grains impose local exchange-bias fields on the adjacent ferromagnetic layer. As a result, the AFM/FM free layer is characterized by a spatially heterogeneous exchange-bias landscape rather than a uniform pinning field. This spatial heterogeneity also provides the physical basis for accessing different intermediate resistance states under different writing current densities, as distinct subsets of antiferromagnetic grains become decoupled from the ferromagnet under the combined effects of Joule heating and spin-orbit torque² and therefore no longer provide effective exchange-bias pinning.

Fig. R37 | Schematic illustration of domain configurations during SOT-induced switching of the device. The ferromagnetic (FM) and antiferromagnetic (AFM) layers are shown for different resistance states during application of the SOT voltage pulse and after relaxation to the stable state.

Next, we describe in more detail how the device resistance states evolve progressively as the amplitude of the writing current is increased. As the writing current increases, a progressive, grain-resolved decoupling process occurs in the AFM/FM free layer under the combined effects of Joule heating and spin-orbit torque^{2,3}, as illustrated by the micromagnetic simulations in Fig. R37. The upper panels (transient dynamics) depict the magnetic configurations of the ferromagnetic and antiferromagnetic layers during current application. In this process, antiferromagnetic grains with lower effective energy barriers are thermally activated and gradually lose their exchange-bias pinning capability, giving rise to regions where exchange bias is suppressed or vanishes.

In these regions where exchange bias is suppressed or vanishes, the ferromagnetic layer can be locally reversed by the current-induced Oersted field, giving rise to a multi-domain configuration. From state 2 to 4, as the current amplitude further increases, the fraction of reversed ferromagnetic domains continues to grow, while the number of antiferromagnetic grains providing effective pinning progressively decreases. This evolution directly reflects a current-dependent redistribution of exchange bias at the grain level. The threshold for AFM decoupling can be expressed in terms of a critical grain volume and applied voltage³:

$$v < v_c = \frac{k_B T \ln(\tau_m/\tau_0)}{K_{AF} \left(1 - \frac{V}{V_{c0}}\right)}$$

where v_c is the critical grain volume at a given temperature, V_{c0} is the threshold voltage at zero temperature, V denotes applied voltage.

After termination of the writing current, the Oersted field vanishes and the device temperature relaxes toward equilibrium. Consequently, the ferromagnetic domain configuration exhibits slight relaxation compared to its state during current application. Simultaneously, the antiferromagnetic grains re-couple with the adjacent ferromagnetic domains. In regions where the ferromagnetic magnetization has been reversed, the exchange bias is re-established with an orientation aligned to the reversed ferromagnetic state, forming a locally reversed exchange-bias field. As highlighted by the red markers in the middle panels of Fig. R37, this reconfigured exchange bias strongly pins the reversed ferromagnetic domains, thereby stabilizing the intermediate magnetic configuration.

Fig. R38 | Schematic R - H curves of the device at different resistance states.

This simulation-based picture is further confirmed by experimental R - H curves measured at different stabilized intermediate resistance states of the EB-MTJ, as shown in Fig. R38. After programming the device into a specific intermediate resistance state, the resistance was measured as a function of the applied magnetic field, from which the effective exchange-bias field was extracted based on the loop shift. As the writing current increases, the extracted exchange-bias

field continuously decreases, approaches zero, and subsequently increases in the opposite direction. This evolution indicates that, as ferromagnetic domain reversal progresses, an increasing number of antiferromagnetic grains undergo exchange-bias reorientation. In the intermediate regime, reversed and unreversed antiferromagnetic grains coexist and partially compensate each other, resulting in a reduced net exchange bias, which eventually reverses and grows in magnitude as the reoriented grains dominate.

In summary, the combination of grain-resolved decoupling, local ferromagnetic reversal, and subsequent re-establishment of exchange bias provides a microscopic mechanism for the emergence of stable intermediate resistance states in our EB-MTJs. This mechanism enables discrete, reproducible multilevel configurations, in contrast to the binary switching behavior of conventional SOT-MTJs with purely ferromagnetic free layers.

To provide a more detailed explanation of the physical mechanism underlying the multilevel behavior of our devices, along with the corresponding simulation and experimental results, and to further elucidate the effects of temperature and bias voltage on the device TMR, we have replaced the original Fig. 2e with Fig. R37. The revised figure enables readers to clearly identify the distinct stages of device switching and the associated changes in both the FM and AFM layers. Correspondingly, we have updated the main text at line 173:

“The grain-size distribution can be described by a log-normal function, which can be expressed as

$$f(v)dv = \frac{1}{\sqrt{2\pi}\sigma v} \exp\left[-\frac{(\ln(v)-\mu)^2}{2\sigma^2}\right]dv \quad (1)$$

where v is the grain size and $\ln(v)$ is normally distributed with a mean value μ and a standard deviation σ . The fit parameters are $\mu = 2.04$ and $\sigma = 0.190$ for the $\text{Ir}_{20}\text{Mn}_{80}$ single layer (yellow), and $\mu = 2.14$ and $\sigma = 0.199$ for the sandwiched structure (green), corresponding to median grain sizes of 7.71 nm and 8.47 nm, respectively.”

And provided a detailed explanation about the exchange-bias reconfiguration at line 189:

“This partial switching originates from the heterogeneous exchange-bias landscape imposed by the AFM layer. Antiferromagnets such as $\text{Ir}_{20}\text{Mn}_{80}$ exhibit a distribution of grain sizes, giving rise to a broad distribution of energy barriers among individual grains. Larger antiferromagnetic grains possess sufficient anisotropy energy to remain magnetically stable at room temperature, with their blocking temperature determined by the thermal activation condition:

$$T_B(v) = K_{AF} * v / [k_B * \ln(\tau_m/\tau_0)]. \quad (2)$$

where K_{AF} is the anisotropy energy density of the antiferromagnet, k_B is the Boltzmann constant, τ_m is the measurement time, and τ_0 is the attempt time. During a 0.4 ns SOT pulse, AFM grains with smaller lateral dimensions undergo pronounced precessional dynamics driven by the combined effects of Joule heating and SOT, leading to their dynamical decoupling from the neighboring FM

domains and the local loss of AFM/FM interfacial exchange coupling, as shown in the upper panels (transient dynamics) of Fig. 2e. In these regions, the local FM domains can be reversed by the combined action of the Oersted field and SOT. The coexistence of such locally reversed domains leads to an intermediate magnetic configuration in which the FM layer adopts a multidomain state. The AFM grains that undergo dynamical decoupling are characterized by

$$v < v_c = \frac{k_B T \ln(\tau_m/\tau_0)}{K_{AF} \left(1 - \frac{V}{V_{c0}}\right)} \quad (3)$$

where v_c is the critical grain volume at a given temperature, V_{c0} is the threshold voltage at zero temperature and V denotes the applied voltage.

Upon current withdrawal, as the temperature decreases, the AFM domains beneath the already reversed FM domains re-couple to the FM domains and re-establish a local exchange bias that follows the switched FM magnetization. As a consequence, the FM multidomain configuration becomes strongly pinned by the reformed exchange bias and is rendered highly stable, as shown in the middle panels of Fig. 2e. This grain-resolved re-coupling process results in a spatially mixed exchange-bias landscape.

As a result, the exchange-bias reconfiguration and the resulting spatially inhomogeneous exchange-bias landscape described above act to stabilize the multidomain configuration of the FM layer. This stabilization directly manifests at the device level as distinct intermediate resistance states of the MTJ, as captured in the bottom panel of Fig. 2e. As the SOT current amplitude increases, a larger fraction of AFM grains undergoes decoupling and reorientation, progressively enhancing the degree of mixed exchange bias and leading to a gradual transition of the device from the AP to the P state (States 2-5). This grain-selective pinning mechanism is fundamentally distinct from conventional SOT-MTJs with purely ferromagnetic free layers, where the absence of such exchange-bias reconfiguration leads to binary switching behavior.”

Furthermore, a detailed discussion of the impact of the DC read current on the device TMR is provided in Section 2 of the Supplementary Information.

[1] Vallejo-Fernandez, G., Fernandez-Outon, L. E. & O’Grady, K. Antiferromagnetic grain volume effects in metallic polycrystalline exchange bias systems. *J. Phys. D: Appl. Phys.* 41, 112001 (2008).

[2] Cai, W. et al. Anatomy of Thermally Interplayed Spin-Orbit Torque Driven Antiferromagnetic Switching. Preprint at <https://doi.org/10.48550/arXiv.2410.13202> (2024).

[3] Du, A. et al. Electrical manipulation and detection of antiferromagnetism in magnetic tunnel junctions. *Nature Electronics* 6, 425–433 (2023).

Q2. How about the impact of process variations? Since the margin between the states is very small, even small variations in material properties or fabrication steps can lead to significant mismatches in device behavior and performance.

Response: We thank the reviewer for raising the important issue of device-to-device uniformity, which is highly relevant for the further optimization of device arrays and for future large-scale integration. Following the reviewer's suggestion, we performed a systematic statistical analysis of device uniformity from three perspectives: (i) the resistance distribution of synaptic devices with elliptical junctions of approximately 80×160 nm, (ii) the SOT switching characteristics, and (iii) the number of accessible resistance states. The corresponding statistical results are summarized in Figs. R39, R40, and R41, respectively.

Fig. R39 | Distribution statistics of P-state resistance and TMR across multiple devices.

Fig. R40 | Distribution of SOT critical switching voltages and corresponding switching curves for multiple devices.

Fig. R41 | Distribution of the number of resistance states across multiple devices.

Note that, under the current laboratory-scale fabrication conditions, the P-state resistance exhibits a certain level of variation, distributed mainly in the range of 12–19 k Ω . In contrast, the TMR shows a much more concentrated distribution, primarily between 75% and 85%, indicating a relatively stable spin-transport performance despite the resistance fluctuation. Meanwhile, the SOT switching characteristics also display moderate device-to-device variation, with an approximately 10.5% difference in the threshold voltage required for complete switching into the P state. Nevertheless, all devices are able to support more than 20 stable resistance states, and the distribution of the number of states is relatively narrow, which already satisfies the basic requirements for multi-level synaptic programming in neuromorphic computing.

We attribute the observed variations mainly to the fact that these devices were fabricated as laboratory-scale samples using manually optimized EBL exposure and etching parameters. As the process conditions are not yet fully standardized, small deviations in the actual device dimensions are unavoidable, and the etching process may also induce slight edge damage. These factors can jointly contribute to the observed spread in device resistance and TMR values¹⁻³.

To further verify that the observed non-uniformity mainly originates from the laboratory-scale fabrication process rather than from intrinsic material or physical limitations, we fabricated an additional batch of circular devices with a diameter of 700 nm using a commercial semiconductor foundry, while keeping the same multilayer stack. Owing to the constraints of the foundry process, the device size is larger than that of the laboratory-fabricated samples; nevertheless, this batch provides a reliable reference for evaluating process-induced uniformity.

Fig. R42 | Device characteristics from foundry fabrication. **a** Statistical distribution of P-state resistance for multiple devices. **b** Statistical distribution of TMR for multiple devices. **c** Representative switching curves of multiple devices.

As shown in Fig. R42, the P-state resistance of the foundry-fabricated devices exhibits a highly concentrated distribution, with the maximum-to-minimum variation being only ~7%. Meanwhile, all devices demonstrate a TMR exceeding 100%, with values mainly distributed between 102% and 112%, corresponding to a standard deviation of only 3.22%. In addition, the

SOT switching curves measured under 50 ns current pulses are significantly more clustered compared with those of the laboratory-fabricated devices, further confirming the excellent device-to-device uniformity achievable under mature and well-controlled fabrication conditions.

These results strongly suggest that the uniformity of EB-MTJ synaptic devices can be substantially improved by transitioning from laboratory-scale fabrication to industrial-grade semiconductor processes. The remaining differences between the two batches are therefore not fundamental, but rather process-related, and are expected to be further mitigated through standardized lithography and etching conditions.

We further note that, although the laboratory-fabricated devices exhibit relatively larger variations, their current performance is already sufficient for the implementation of the fully EB-MTJ-based CSNN demonstrated in this work. To quantitatively assess the impact of device non-uniformity on system-level performance, we incorporated the experimentally extracted distributions of resistance states and state-to-state variations into network-level simulations.

Fig. R43 | Impact of device variations on network accuracy. **a** Classification accuracy as a function of the number of synaptic resistance states. When the number of states exceeds 15, the network accuracy remains above 94%. **b** Classification accuracy as a function of the write error associated with each synaptic resistance state.

Fig. R43a illustrates the impact of the number of synaptic resistance states on the gesture-recognition accuracy of the full EB-MTJ-based CSNN proposed in the main text. When the number of resistance states is 15, the network achieves a maximum accuracy of 94.8%, which is only 1.4% lower than that obtained with 25 resistance states (96.2%), indicating that the network performance remains largely preserved even with a slightly reduced state number. In addition, considering that write operations based on SOT voltage pulses may inevitably introduce write errors, such as transitions to neighboring resistance states, we further analyze the influence of synaptic write error on network accuracy, as shown in Fig. R43b. Notably, even at a write error rate of 10%, the network maintains stable operation and high recognition accuracy. These results

collectively demonstrate that the proposed EB-MTJ-based neuromorphic architecture exhibits strong tolerance to device non-idealities and intrinsic variations.

Finally, we would like to emphasize that the primary objective of this study is not to demonstrate large-scale device integration or state-of-the-art uniformity, but rather to introduce a monolithically integrated EB-MTJ-based CSNN architecture capable of achieving a hand-gesture recognition accuracy exceeding 96%. Importantly, the experimentally demonstrated device performance already meets the accuracy and reliability requirements of the proposed CSNN. With further improvements in device uniformity enabled by mature foundry fabrication, the system-level performance is expected to be further enhanced, rather than fundamentally limited.

In line 326 of the revised manuscript, we have revised the main text to indicate that the impact of device variability on network accuracy is analyzed in SI, as follows:

“As shown in Fig. 4c, the hybrid scheme achieved a test accuracy over 96% after 100 training epochs, outperforming pure BP and pure STDP by 1.5 and 11.7 percentage points, respectively. After accounting for device-to-device variability, the proposed scheme maintains an accuracy above 94%, as described in Section 12 of the SI.”

Section 12 of the Supplementary Information provides a statistical analysis of the variability across existing devices, presents the variability observed in foundry-fabricated devices, and examines how these variations affect network accuracy.

[1] Zhao, W. et al. Failure Analysis in Magnetic Tunnel Junction Nanopillar with Interfacial Perpendicular Magnetic Anisotropy. *Materials* 9, (2016).

[2] Ziaur Rahaman, Sk. et al. Process-induced magnetic tunnel junction damage and its recovery for the development of spin-orbit torque magnetic random access memory. *Journal of Magnetism and Magnetic Materials* 565, 170296 (2023).

[3] Zhang, L., Zhou, J., Li, H., Shen, L. & Feng, Y. P. Recent progress and challenges in magnetic tunnel junctions with 2D materials for spintronic applications. *Appl. Phys. Rev.* 8, 021308 (2021).

Q3. In Figure-1(a), the authors present an SOT device, whereas in Figure-1(b), the switching mechanism corresponds to an STT device. However, SOT and STT are two fundamentally different technologies with distinct switching mechanisms. Please clarify this inconsistency. Additionally, why an in-plane MRAM structure considered when perpendicular MRAM is known to offer better switching efficiency (in terms of faster and low-energy switching)?

Response: We thank the reviewer for pointing out the unclear description in our manuscript. Throughout this work, all experiments are conducted using the SOT device configuration shown in Fig. 1a. During the write operation, two voltage pulses with identical amplitude but opposite polarity are simultaneously applied to the two ends of the bottom electrode, ensuring that the

electric potential at the center of the bottom electrode remains zero. Consequently, no voltage drop is applied across the MTJ stack, and the switching process is purely SOT-driven and does not involve spin-transfer torque (STT).

Importantly, the measurement shown in Fig. 1b probes the magnetic-layer response of the SOT-MTJ under an external magnetic field by monitoring the device resistance using a small sensing current while sweeping the applied field, rather than representing STT-induced switching. In the original schematic, the bottom electrode was omitted to emphasize the magnetization configurations of the reference layer, free layer, and antiferromagnetic layer. We acknowledge that this simplification may have caused confusion. To avoid ambiguity, Fig. 1b has been revised in the revised manuscript to explicitly include the bottom electrode and clarify the device configuration, as shown in Fig. R44.

Fig. R44 | Magnetic hysteresis loops of the device.

Our device is based on an EB-MRAM architecture, in which an interfacial exchange coupling exists between the bottom AFM layer and the free layer. This exchange coupling pins the magnetization of the ferromagnetic layer along a fixed in-plane direction defined by the AFM layer¹⁻³. Importantly, the AFM layer is insensitive to external magnetic fields⁴. As a result, although the magnetization of the free layer may rotate or cant under an applied magnetic field, it always relaxes back to its initial stable configuration once the external field is removed, driven by the exchange-bias field from the AFM layer⁵. Consequently, in contrast to conventional SOT-MRAM, which may exhibit two stable resistance states at zero field, the EB-MRAM device possesses only a single stable resistance state in the absence of an external magnetic field. As illustrated by the black (red) curve in Fig. R41, we define this zero-field state as the \mathbf{n}^- (\mathbf{n}^+) state. Because the AFM layer is not affected by the external magnetic field, the device always relaxes back to the AP (P) state at zero field. Importantly, the state of the AFM layer—and thus the \mathbf{n}^- or \mathbf{n}^+ state of the device—can only be written by SOT current, rather than by an external magnetic field.

In the revised manuscript, Fig. 1b has been revised to explicitly include the bottom electrode and clarify the device configuration. The updated data are shown in the revised Fig. 1b of the main text and are also provided here as Fig. R44. In addition, the main text at line 116 has been revised to clarify the writing of the AFM state:

“Notably, the n^- and n^+ configurations are preset electrically by SOT current prior to the field sweep in Fig. 1b.”

We also appreciate the reviewer’s comment regarding the advantages of perpendicular MRAM in terms of switching speed, energy efficiency, and scalability. We fully agree that perpendicular MTJs represent an important and highly successful technological platform. However, the primary objective of this work is to realize multistate synaptic functionality, rather than to optimize switching performance. Achieving a large number of stable and reproducible intermediate resistance states in conventional perpendicular MTJs remains challenging⁶⁻⁹, as previously reported studies typically demonstrate fewer than ~20 distinguishable states, with stability and scalability being nontrivial.

In this context, FM/AFM-based in-plane MTJs provide an alternative route for multilevel operation. The interfacial exchange-bias effect can stabilize partial magnetization reversal and intermediate magnetic configurations, which is favorable for multistate resistance storage¹⁰. In perpendicular systems, the exchange-bias field achievable at FM/AFM interfaces is often comparable to or smaller than the coercive field of the ferromagnetic layer, limiting its effectiveness in pinning intermediate states^{11,12}. By contrast, the exchange bias in in-plane EB-MTJs can more effectively stabilize intermediate magnetization states, enabling multilevel operation even at submicron device dimensions.

Finally, the in-plane geometry naturally satisfies the condition that the magnetization direction is perpendicular to the SOT current, enabling intrinsic field-free switching without auxiliary symmetry-breaking schemes, which further supports reliable synaptic operation.

[1] Peng, S. et al. Exchange bias switching in an antiferromagnet/ferromagnet bilayer driven by spin-orbit torque. *Nat Electron* 3, 757–764 (2020).

[2] Zhu, D. Q. et al. First demonstration of three terminal MRAM devices with immunity to magnetic fields and 10 ns field free switching by electrical manipulation of exchange bias. in 2021 IEEE International Electron Devices Meeting (IEDM) 17.5.1-17.5.4 (2021). doi:10.1109/IEDM19574.2021.9720599.

[3] Cai, W. et al. Anatomy of Thermally Interplayed Spin-Orbit Torque Driven Antiferromagnetic Switching. Preprint at <https://doi.org/10.48550/arXiv.2410.13202> (2024).

- [4] Néel, M. L. Propriétés magnétiques des ferrites ; ferrimagnétisme et antiferromagnétisme. *Ann. Phys.* 12, 137–198 (1948).
- [5] Du, A. et al. Electrical manipulation and detection of antiferromagnetism in magnetic tunnel junctions. *Nature Electronics* 6, 425–433 (2023).
- [6] Zhang, D. et al. All Spin Artificial Neural Networks Based on Compound Spintronic Synapse and Neuron. *IEEE Transactions on Biomedical Circuits and Systems* 10, 828–836 (2016).
- [7] Zhang, X. et al. Spin-Torque Memristors Based on Perpendicular Magnetic Tunnel Junctions for Neuromorphic Computing. *Advanced Science* 8, 2004645 (2021).
- [8] Kurenkov, A. et al. Artificial Neuron and Synapse Realized in an Antiferromagnet/Ferromagnet Heterostructure Using Dynamics of Spin–Orbit Torque Switching. *Advanced Materials* 31, 1900636 (2019).
- [9] Liu, L. et al. Domain wall magnetic tunnel junction-based artificial synapses and neurons for all-spin neuromorphic hardware. *Nature Communications* 15, 4534 (2024).
- [10] Nogués, J., Lederman, D., Moran, T. J. & Schuller, I. K. Positive Exchange Bias in FeF₂-Fe Bilayers. *Phys. Rev. Lett.* 76, 4624–4627 (1996).
- [11] Tang, X. L., Zhang, H. W., Su, H., Jing, Y. L. & Zhong, Z. Y. Spin-transfer effect and independence of coercivity and exchange bias in a layered ferromagnet/antiferromagnet system. *Phys. Rev. B* 81, 052401 (2010).
- [12] Radu, F., Abrudan, R., Radu, I., Schmitz, D. & Zabel, H. Perpendicular exchange bias in ferrimagnetic spin valves. *Nat Commun* 3, 715 (2012).

Q4. How about the retention capability of these devices? It is also not clear what cost is associated with the Joule heating effect in these EB-based MTJ devices, even though the authors claim energy-efficient, ultra-fast switching of 190 fJ/bit.

Response: We thank the reviewer for pointing out the aspects related to retention that required clarification. We evaluate the retention properties of our devices using two approaches.

First, we estimate the retention based on an ideal single-domain magnetic model^{1,2}. For the circular neuron devices, the ferromagnetic layer exhibits two lowest-energy stable states, namely the AP and P states (for the synaptic devices, the midpoint of the TMR can be taken as the boundary). These two stable states are separated by an energy barrier E_b . Assuming only thermally induced single-domain magnetization reversal, the average switching time τ between the two states follows the Néel–Brown law³:

$$\tau = \tau_0 \exp\left(\frac{E_b}{k_B T}\right)$$

where τ_0 is the attempt time (typically 10^{-9} s), k_B is the Boltzmann constant, T is the temperature. The quantity $\Delta = E_b/k_B T$ is commonly defined as the thermal stability factor.

Next, for conventional SOT-driven MTJ switching, the switching behavior can be divided into two regimes depending on the pulse width, as reported in previous studies⁴⁻⁶. As illustrated in Fig. R45, when the pulse width is short, the switching occurs in a non-fully thermally activated regime, where the critical switching voltage shows an approximately inverse dependence on pulse width. In contrast, for longer pulse widths, switching occurs in the thermally activated regime, where the critical switching voltage exhibits a linear dependence on the pulse width. In this regime, the relationship between the critical switching voltage V_C and the pulse width τ can be expressed as^{7,8}

$$V_C = V_{C0} \left[1 - \frac{k_B T}{E_b} \ln\left(\frac{\tau}{\tau_0}\right) \right]$$

where V_{C0} is the critical switching voltage in the absence of thermal activation. By fitting this relation, an estimate of the thermal stability factor can be obtained, which in turn allows the retention time to be inferred. To ensure that the switching process is dominated by thermal activation, we performed the fitting using critical switching voltages corresponding to pulse widths longer than 10 μ s. From this analysis, the extracted **thermal stability factor Δ is approximately 113.3**, which is well above the typical requirement for ten-year data retention at room temperature.

Fig. R45 | Critical switching voltage as a function of pulse width, with fitting in the long-pulse-width regime using a thermal activation model.

Moreover, to complement the analytical estimation, we evaluated the retention using a thermally accelerated aging test. Figure R46a shows the switching behavior of the device at an elevated temperature of 470 K using 100-ns SOT pulses. Then, the device was programmed into an intermediate resistance state under these conditions, and the resistance was monitored continuously. As shown in Fig. R46b, the resistance state remains stable for more than 2650 minutes (>44 hours) at 470 K, indicating good thermal robustness of the device, indicating a retention time higher than 10 years at room temperature (300 K). These results further indicate that the non-volatile resistance state of our EB-MTJ is robust against thermal perturbations.

Fig. R46 | Device retention at high temperature. a R–V curves measured at 470 K. **b** Temporal evolution of the intermediate resistance state at 470 K.

To further clarify the retention characteristics and thermal stability of the device, we have added the following description in the main text at line 161:

“Specifically, the EB-MTJ exhibits a thermal stability factor exceeding 100, corresponding to a retention time exceeding 10 years, as detailed in Section 7 of the SI.” The corresponding detailed discussion has been included in Section 7 of the Supplementary Information.

In addition, we additionally appreciate the reviewer for highlighting the important aspect related to the impact of Joule heating effects. We address the potential thermal crosstalk from two perspectives: individual device separation and system-level heating effects.

First, at the device level, thermal crosstalk is highly dependent on the physical spacing and thermal dissipation characteristics of the integrated array. In our group’s previous publication (Jiang et al.⁹), cross-sectional TEM images (Fig. R47b) of a 128 Kb SOT-MRAM array show that neighboring MTJ cells are physically separated by more than 5 μm , with multiple interleaved metal routing layers and insulating oxide layers in between. As shown in Fig. R48, we have used finite element COMSOL simulations to further confirm that, at such separations, the temperature rise in adjacent MTJs induced by a firing event is negligible, resulting in no significant thermal crosstalk at the device scale. We set a heating source with dimensions of 100 nm (length) \times 300 nm (width) \times 8 nm (thickness), through which a constant current density is applied. The temperature distribution as a function of distance from the device was systematically characterized. Our results reveal an exponential decay of temperature increase with distance, with the temperature rise falling below 5 K at distances exceeding 500 nm. In actual operation, nanosecond-scale current pulses were employed instead of DC signals. Combined with the inherent spike sparsity in spiking neural networks, the practical temperature increase is expected to be significantly lower than 5 K and thus negligible in practice cases.

Second, at the system level, we analyzed the thermal impact based on the overall spike activity during network operation. For the training process, approximately 1,146,880 spikes were generated, with each spike consuming 190 fJ. Assuming an average spike interval of 1 ns, and a chip volume corresponding to 512 MTJ cells arranged over an area of $6\ \mu\text{m} \times 6\ \mu\text{m}$ per MTJ, we estimate the energy input and subsequent temperature rise. Using the specific heat capacity of SiO_2 as a reference ($\sim 0.7\ \text{J/g/K}$) and standard density values, the calculated average temperature rise across the chip remains well below 4 K, even under continuous operation.

Fig. R47 | a The pictures of the 128 Kb SOT-MRAM chip package. b TEM images of a single device in the chip⁹

Fig. R48 | a COMSOL heat simulation of MTJ. b One-dimensional temperature distribution.

Thus, both from the individual device perspective and the overall system thermal budget, thermal crosstalk and overheating are negligible under the current design and operating conditions.

[1] Thomas, L., Jan, G., Le, S. & Wang, P.-K. Quantifying data retention of perpendicular spin-transfer-torque magnetic random access memory chips using an effective thermal stability factor method. *Appl. Phys. Lett.* 106, 162402 (2015).

- [2] Wernsdorfer, W. et al. Experimental Evidence of the Néel-Brown Model of Magnetization Reversal. *Phys. Rev. Lett.* 78, 1791–1794 (1997).
- [3] Brown, W. F. Thermal Fluctuations of a Single-Domain Particle. *Phys. Rev.* 130, 1677–1686 (1963).
- [4] Du, A. et al. Electrical manipulation and detection of antiferromagnetism in magnetic tunnel junctions. *Nature Electronics* 6, 425–433 (2023).
- [5] Krizakova, V. et al. Tailoring the switching efficiency of magnetic tunnel junctions by the fieldlike spin-orbit torque. *Phys. Rev. Applied* 18, 044070 (2022).
- [6] Cubukcu, M. et al. Ultra-Fast Perpendicular Spin–Orbit Torque MRAM. *IEEE Transactions on Magnetics* 54, 1–4 (2018).
- [7] Ikeda, S. et al. A perpendicular-anisotropy CoFeB–MgO magnetic tunnel junction. *Nature Mater* 9, 721–724 (2010).
- [8] Rizzo, N. D. et al. Thermally activated magnetization reversal in submicron magnetic tunnel junctions for magnetoresistive random access memory. *Appl. Phys. Lett.* 80, 2335–2337 (2002).
- [9] Jiang, C. et al. Demonstration of 128 Kb SOT-MRAM Chip with 5 ns Write and 15 ns Read Speed, High Endurance Over 10¹⁰ and Low ECC-on Bit Error Rate. in *2024 IEEE International Electron Devices Meeting (IEDM)* 1–4 (2024). doi:10.1109/IEDM50854.2024.10873510.

Q5. Authors have specified that they employed a hybrid back-propagation and STDP approach, where they claim back-propagation alleviates the vanishing plasticity problem, while STDP introduces biologically inspired local learning. However, the working principles of these two learning approaches are completely different, means difficult to integrate them. Secondly, it is not clear how the authors have dealt with the gradient mismatch problem in this case.

Response: We thank the reviewer for this important comment. We agree that naively mixing STDP and back-propagation (BP) on the same synapses can be problematic, and the “gradient mismatch” issue must be handled carefully in SNN training. In our implementation, we adopt a widely used and well-established hybrid strategy in Spiking Jelly¹: STDP and gradient descent are assigned to different parameter groups and updated by different optimizers, rather than being simultaneously applied to the same weight set.

During the training process, we use surrogate-gradient BP to train the convolutional synapses (device-mapped synaptic layers), while the last full connected layer is trained by STDP. Following the workflow, we create an `STDPLearner` instance for each STDP-trained layer, split network parameters into two disjoint sets (STDP parameters vs. gradient-descent parameters), and use two optimizers (e.g., SGD for STDP layers and Adam for BP layers). During each training iteration,

we first perform a standard forward pass and compute the loss, then call `loss.backward()` to compute gradients.

A key practical issue is that `loss.backward()` computes gradients for all parameters, including those intended to be updated by STDP. To avoid conflicting updates (and thus reduce mismatch between BP and local plasticity), we explicitly zero out the gradients of STDP-trained parameters after backpropagation, and then let `STDPLearner` write the STDP update into the gradient buffer via `stdp_learner.step(on_grad=True)`. Finally, the two optimizers update their corresponding parameter sets (`optimizer_gd.step()` for BP-trained layers and `optimizer_stdp.step()` for STDP-trained layers). This procedure ensures that BP-derived gradients do not drive the STDP layers, and conversely STDP updates do not interfere with the BP parameter subset.

For the BP-trained layers, the remaining gradient mismatch associated with the non-differentiable spike function is handled using the standard surrogate-gradient method, where the Heaviside function is used in the forward pass and a smooth surrogate derivative is used for the backward pass. This is the only approximation involved in the BP path. Additionally, we reset the internal traces/monitors of `STDPLearner` at the end of each batch to prevent memory growth.

In the revised manuscript, we added the following description at line 322:

“To prevent update conflicts and mitigate gradient mismatch, we decouple the learning paths by applying BP only to the BP-trained parameter group with surrogate gradients for spikes and applying STDP only to the STDP-trained layer via a separate local update/optimizer after clearing BP gradients on that layer.”

[1] Fang, Wei, et al. "Spikingjelly: An open-source machine learning infrastructure platform for spike-based intelligence." *Science Advances* 9.40 (2023): eadi1480.

Q6. A general comments, authors are claiming high TMR, but >70% is not high enough, especially for the neuromorphic applications.

Response: We thank the reviewer for pointing out the inappropriate wording in our original manuscript. This comment is very helpful for improving the accuracy and clarity of our presentation. We fully agree that a TMR value of ~70% should not be described as “high”. Instead, we refer to it as moderate, which is sufficient to clearly resolve multiple resistance states in the synaptic device.

The moderate TMR observed in this work mainly arises from practical constraints associated with device fabrication and measurement conditions. First, the devices reported here were fabricated using laboratory-scale processes, as the primary goal of this study was functional demonstration and physical validation rather than process optimization. For small-sized devices, edge effects and etching-induced damage become more pronounced, which can degrade interfacial

quality and reduce the measured TMR^{1,2}. Second, in order to reliably probe nanosecond-scale switching dynamics, the RA product of the devices must remain relatively low. Consequently, a thinner MgO barrier was employed, which weakens coherent tunneling filtering and leads to a reduced TMR compared to devices optimized purely for memory performance³⁻⁵.

Fig. R49 | Statistical distribution of TMR for multiple devices fabricated by the foundry.

It is important to note that these limitations are not fundamental to the EB-MTJ platform itself. As shown in Fig. R49, when devices with larger lateral dimensions are fabricated using industrial-grade processes, the TMR can be significantly improved to values exceeding 100%. Moreover, in application scenarios where ultrafast switching dynamics are not required, a thicker MgO barrier can be adopted to enhance coherent tunneling and further increase the TMR. While MRAM generally exhibits a smaller resistance on/off ratio than some other emerging memory technologies, MRAM-based synaptic devices can still achieve switching ratios above 100%, which is sufficient to support neuromorphic computing functionalities. Importantly, MRAM offers multiple current-driven switching mechanisms, enabling diverse routes to emulate different characteristics of biological neural networks, without relying solely on a large resistance contrast⁶⁻⁸.

In the revised manuscript, we added the following description at line 107:

“MTJs with diverse dimensions and aspect ratios are fabricated, consistently exhibiting TMR values exceeding 70%, which are sufficient for reliable discrimination of multiple resistance states, ...”

[1] Zhao, W. et al. Failure Analysis in Magnetic Tunnel Junction Nanopillar with Interfacial Perpendicular Magnetic Anisotropy. *Materials* 9, (2016).

[2] Ziaur Rahaman, Sk. et al. Process-induced magnetic tunnel junction damage and its recovery for the development of spin-orbit torque magnetic random access memory. *Journal of Magnetism and Magnetic Materials* 565, 170296 (2023).

[3] Butler, W. H., Zhang, X.-G., Schulthess, T. C. & MacLaren, J. M. Spin-dependent tunneling conductance of Fe|MgO|Fe sandwiches. *Phys. Rev. B* 63, 054416 (2001).

- [4] Yuasa, S. & Djayaprawira, D. D. Giant tunnel magnetoresistance in magnetic tunnel junctions with a crystalline MgO(0 0 1) barrier. *J. Phys. D: Appl. Phys.* 40, R337 (2007).
- [5] Hayakawa, J., Ikeda, S., Matsukura, F., Takahashi, H. & Ohno, H. Dependence of Giant Tunnel Magnetoresistance of Sputtered CoFeB/MgO/CoFeB Magnetic Tunnel Junctions on MgO Barrier Thickness and Annealing Temperature. *Jpn. J. Appl. Phys.* 44, L587 (2005).
- [6] Zhang, X. et al. Spin-Torque Memristors Based on Perpendicular Magnetic Tunnel Junctions for Neuromorphic Computing. *Adv. Sci.* 8, 2004645 (2021).
- [7] Zhou, J. et al. Spin–Orbit Torque-Induced Domain Nucleation for Neuromorphic Computing. *Adv. Mater.* 33, 2103672 (2021).
- [8] Zhu, D. et al. Thermally Driven Leaky-Integrate-and-Fire Spintronic Neurons with Stray-Field-Enabled Self-Reset for Neuromorphic Computing. *IEEE Electron Device Lett.* 1–1 (2025).

Response to referees' letter:

We sincerely thank the reviewers for their careful evaluation of our manuscript and for their constructive and encouraging feedback. We are grateful for their positive assessment of our work and their recognition of its significance. We have addressed the remaining comment and revised the manuscript accordingly. All changes are marked in blue. Our point-by-point responses are provided below.

Response To the Reviewer #1

The authors have responded very well to my questions. Although I am not an expert in magnetic devices, from the perspective of a researcher working on neuromorphic devices, I find the implementation of second-order synaptic characteristics—enabling local learning rules to be realized intrinsically at the device level—to be particularly impressive. While the application of the neuron is somewhat limited due to the need for additional peripheral circuitry, the second-order characteristics are implemented in a stable manner, and the simulations based on these characteristics are conducted at a convincing and reasonable level.

Response: We thank the reviewer for the constructive and encouraging comments. The recognition of the robustness of the second-order synaptic characteristics and the convincing level of the associated simulations is greatly appreciated. We also appreciate the reviewer's comment regarding the current reliance on peripheral circuitry for neuronal operation. Looking ahead, efforts will focus on optimizing the neuron–circuit interface and pursuing tighter integration of neuronal functionalities into the device platform, with the aim of reducing peripheral overhead and progressing toward a more compact, fully integrated “true” spiking neuron.

Response To the Reviewer #2

The authors have carefully addressed each point raised with modified or additional text, additional data, and modified or additional plots. I recommend acceptance.

Response: We sincerely thank the reviewer for the careful and thoughtful evaluation of our manuscript. The suggestions and comments provided throughout the review process have greatly contributed to improving the clarity, rigor, and overall quality of our work. We also greatly appreciate the recognition of our efforts and the recommendation for acceptance. Your feedback is highly valued and reinforces the significance of our study.

Response To the Reviewer #3

Q1. The authors have specified only the TMR values. However, the measured TMR depends on both the read voltage and temperature. Therefore, the corresponding bias voltage and temperature at which the TMR was measured need to be specified.

Response: We thank the reviewer for pointing out the lack of clarity regarding the measurement conditions, which has helped improve the rigor of the manuscript. Except for the retention tests performed under varying temperatures, all tunneling magnetoresistance (TMR) measurements were conducted at room temperature (300 K). The measurements were carried out using a Keithley 6221A current source and a Keithley 2182A voltmeter. A DC bias current of 1 μ A was applied, and the voltage across the device was recorded to extract the resistance. As the neuron device operates via bipolar switching and exhibits a large resistance contrast between the two states, the influence of bias voltage and ambient temperature is negligible. Therefore, our analysis focuses on the resistance readout of the synaptic devices.

For synaptic devices at different intermediate resistance states, the corresponding read bias voltage ranges from approximately 13 mV to 33 mV. Figure R1 shows the dependence of TMR on the read voltage at a fixed resistance state. As highlighted by the red dashed box, the TMR variation remains below 0.8% within the 13–33 mV bias range. This variation is negligible for distinguishing different resistance states during synaptic switching, indicating that the measurement temperature and corresponding bias voltage have limited influence on the extracted TMR values.

Figure R1 | Dependence of TMR on read voltage for the synaptic device at room temperature (300 K).

To provide a detailed description of the measurement conditions for TMR, we have added the following statement in the revised manuscript at line 352:

“Except for the retention tests performed under varying temperatures, all resistance measurements were conducted at room temperature (300 K).”

This manuscript reports the development of memristive synapses and leaky-integrate-and-fire (LIF) neurons based on nanoscale exchange-bias magnetic tunnel junctions (EB-MTJs). The authors appear to utilize heat dynamics to realize synaptic learning behavior akin to so-called second-order synapses. The neuron functionality is also primarily based on thermal effects. While the demonstration of second-order synaptic behavior in EB-MTJ devices is intriguing, the mechanistic explanation remains insufficient, and for the neuron device it is unclear whether genuine neuronal operation has been achieved. Some specific questions are as follows:

Q1. The manuscript reports that elliptical EB-MTJ devices exhibit gradual, analog-like switching while circular devices show abrupt binary transitions. This distinction is crucial for assigning synaptic versus neuronal roles. However, the physical origin of this geometry-dependent behavior is not sufficiently explained. Could the authors provide a more detailed discussion of how device shape (aspect ratio, demagnetizing fields, domain wall nucleation sites) influences the spin-orbit torque efficiency and switching characteristics? Is the observed difference reproducible across a wide range of device sizes and aspect ratios?

Q2. Throughout the manuscript, the authors rely on very short pulse durations (as low as 0.4 ns) and Δt timing to demonstrate both synaptic plasticity and neuronal dynamics. However, the actual pulse waveforms are not shown. Providing oscilloscope traces of the applied voltage/current pulses would significantly aid interpretation of the experiments, allowing readers to assess pulse fidelity, rise/fall times, and possible distortions at short Δt .

Q3. In Fig. 3, the manuscript demonstrates LIF neuron characteristics, describing a “firing” event. In biological neurons, firing typically refers to the generation of an action potential or voltage spike. In the proposed EB-MTJ neuron, however, it appears that firing corresponds to a resistance state transition rather than a transient voltage output. Could the authors clarify how “firing” is defined in their device? Does this resistance change directly translate to a measurable voltage/current spike suitable for network-level signaling, or is additional circuitry required to emulate a biological-like spiking output?

Q4. In biological neurons, firing is followed by an automatic return to the resting potential, providing intrinsic self-reset functionality. For the EB-MTJ neurons presented in this work, it is not clear whether the device spontaneously relaxes back to its initial state after a firing event, or whether external intervention is required. Do the EB-MTJ neurons exhibit genuine self-reset behavior, and if so, what is the underlying mechanism that distinguishes them from EB-MTJ synapses, which maintain non-volatile resistance states? If additional circuitry or stimuli are required for reset, how does this affect the scalability and efficiency of the proposed neuromorphic system?

Q5. For a device to function as a practical spiking neuron, its firing event must generate an

output signal strong enough to drive subsequent synapses in a network. Can the proposed neuron device directly provide the necessary voltage or current pulse to activate downstream EB-MTJ synapses?

Q6. The simulation section lacks sufficient detail to assess how faithfully device-level dynamics were incorporated into the network model. Since the EB-MTJ operation relies heavily on heat-assisted switching, one would expect not only STDP but also other forms of learning dynamics (e.g., BCM-type plasticity) to emerge from the thermal processes.

Could the authors clarify whether their simulations include heat dynamics beyond simple STDP fitting? Was the possibility of BCM-like or other second-order learning rules considered, and if so, how might they affect network-level behavior?